# Why Less is More (Sometimes): A Theory of Data Curation

**Elvis Dohmatob**
Concordia University & Meta FAIR & Mila Institute
`elvis.dohmatob@concordia.ca`

**Mohammad Pezeshki**
Meta FAIR
`mpezeshki@meta.com`

**Reyhane Askari-Hemmat**
Meta FAIR
`reyhaneaskari@meta.com`

## Abstract

This paper introduces a theoretical framework to resolve a central paradox in modern machine learning: When is it better to use less data? This question has become critical as classical scaling laws suggesting "more is more" (Sun et al., 2025) are challenged by methods like LIMO ("less is more") and s1 (Ye et al., 2025; Muenighoff et al., 2025), which achieve superior performance with small, aggressively curated datasets. Here, we study data curation strategies where an imperfect oracle selects the training examples according to their difficulty and correctness. Our results provide exact scaling law curves for test error under both label-agnostic and label-aware curation rules, revealing when and why keeping only a subset of data can improve generalization. In contrast to classical scaling laws, we show that under certain conditions, small curated datasets can outperform full datasets, and we provide analytical conditions for this by deriving precise phase transition curves tied to data size and quality. We validate these theoretical claims with empirical results on ImageNet, confirming our predictions about when curation improves accuracy and can even mitigate model collapse. Furthermore, our framework provides a principled explanation for the contradictory curation strategies recently observed in LLM mathematical reasoning.

## 1 Introduction

Despite remarkable advances in large language models (LLMs) and other foundation models, training them remains highly inefficient, often requiring hundreds of billions of tokens. A key reason lies in how training data is used: standard loss functions treat all examples equally, regardless of their informativeness. Yet not all data points contribute equally to learning; while some accelerate progress, others are redundant or even detrimental (Sorscher et al., 2022). This inefficiency motivates the exploration of principled data curation strategies.

Recent empirical successes highlight the promise of aggressive data curation. Methods such as LIMO (Less Is More) (Ye et al., 2025) and s1 (Muennighoff et al., 2025) show that curating compact sets of valid and challenging examples can dramatically improve reasoning performance, often with a fraction of the original data. These results stand in contrast to the traditional scaling law perspective (Kaplan et al., 2020; Hoffmann et al., 2022), which suggests that simply increasing dataset size should monotonically improve generalization. The apparent contradiction between "less is more" and "more is more" (Sun et al., 2025) raises a fundamental question: under what conditions does data curation help, and when does full-data training remain optimal?

In this work, our goal is not to propose another heuristic curation method, but rather to build a principled theoretical framework that explains why and when such strategies succeed. We analyze high-dimensional binary classification under pruning oracles that filter examples based on difficulty and correctness. Our theory provides exact scaling laws for test error, revealing sharp phase transitions tied to dataset size, label quality, and oracle reliability. These results establish conditions under which keeping only the hardest or easiest examples outperforms training on the full dataset.

Crucially, we show how strategic curation can mitigate *model collapse* (Shumailov et al., 2024; Dohmatob et al., 2024a), where iterative self-training on noisy or synthetic data leads to catastrophic degradation.

**Main Contributions:**

- We develop a precise theoretical framework for data curation in high-dimensional learning, deriving exact scaling laws that characterize the effect of data pruning on generalization.

- We demonstrate that, under realistic compute or label-quality constraints, strategically pruned datasets can outperform full datasets, thereby bending classical scaling laws.

- We empirically confirm our theoretical predictions on ImageNet and connect them to recent large-scale results in LLM reasoning, providing a rigorous justification for why methods like LIMO and s1 succeed.

- We show analytically that data curation can avert model collapse under label shift, establishing phase boundaries where uncurated training diverges while curated training remains stable.

Together, these results reframe data curation not as a heuristic preprocessing step, but as a principled tool for stable and efficient learning.

## 2 SETUP FOR THEORETICAL ANALYSIS

To formally analyze when "less is more" versus when "more is more", we must first establish a precise mathematical setting, which is rich enough to capture the complexity of the problem, but simple enough to be analytically tractable. This section defines our data generation process, the model we analyze, and, most importantly, the key quantities that will allow us to distinguish between different learning regimes: the **quality of the data generator** and the **quality of the pruning oracle**.

### 2.1 DATA, MODEL, AND ASSUMPTIONS

**Data Distributions.** Let $P_{w,A}$ denote the probability distribution on $\mathbb{R}^d \times \mathbb{R}$ given by:

$$(x, y) \sim P_{w,A} \quad \text{iff} \quad x \sim \mathcal{N}(0, A), \ y = \text{sign}(x^\top w). \tag{1}$$

The training dataset consists of $n$ i.i.d. pairs $(x_i, y_i)$ from a distribution $P_g = P_{w_g, C_g}$, where $w_g \in \mathbb{R}^d$ and $C_g \in \mathbb{R}^{d \times d}$ are the weights/labeling vector and the covariance matrix for the generative distribution (the "generator"). The true test data distribution is, however, $P_* = P_{w_*, \Sigma}$, where $w_* \in \mathbb{R}^d$ and $\Sigma \in \mathbb{R}^{d \times d}$ are the true weights and covariance. In general, we consider $w_g \neq w_*$ (i.e., label shift) and $C_g \neq \Sigma$ (i.e., covariate shift).

**The Model.** Consider a vector $\hat{w} \in \mathbb{R}^d$ defined as the solution to the convex optimization problem:

$$\text{minimize } \frac{1}{n} \sum_{i=1}^{n} p_i \ell(x_i^\top w; y_i) + \frac{\lambda}{2} \|w\|^2, \text{ over } w \in \mathbb{R}^d. \tag{2}$$

Here, $\ell(z; y) := (z - y)^2/2$ is the squared L2 loss, $\lambda > 0$ is a regularization parameter, and $p_i \in \{0, 1\}$ indicates if an example is kept. The downstream classifier is $x \mapsto \text{sign}(x^\top \hat{w})$. The first-order condition for optimality in Eqn (2) gives the solution:

$$\hat{w} = RX^\top DY/n, \quad \text{with} \quad R := (S + \lambda I_d)^{-1} \text{ and } S := X^\top DX/n, \tag{3}$$

where $X \in \mathbb{R}^{n \times d}$ is the design matrix, $Y \in \mathbb{R}^n$ is the label vector, and $D$ is a diagonal matrix with $D_{ii} := p_i$, indicating which examples are present.

**Object of Study: High-Dimensional Test Error** Our goal is to characterize the classification test error, $E_{\text{test}}(\hat{w}) := \mathbb{P}(\text{sign}(x^\top \hat{w}) \neq y)$, in the high-dimensional proportionate scaling limit:

$$n, d \to \infty, \quad d/n \to \phi \in (0, \infty). \tag{4}$$

For simplicity of presentation of our main theoretical results and insights, we limit the analysis to the isotropic setting where the covariance matrices are identity matrices, i.e., $C_g = \Sigma = I_d$. More general results are deferred to the appendix. Thus, our focus here is on label shift, where the labels from the generator $P_g$ might deviate from the ground-truth labels from $P_*$.

## 2.2 DATA CURATION RULES

**Label-Agnostic Curation.** First, we consider a setting where an example $(x_i, y_i)$ is retained based only on its features $x_i$, via a pruning function $q : \mathbb{R} \to \{0, 1\}$ and an oracle pruning vector $w_o \in \mathbb{R}^d$:

$$p_i = q(x_i^\top w_o). \tag{5}$$

This rule uses the function $q$ to select examples based on their projection onto the oracle vector $w_o$. For instance, common strategies like "keep easy" and "keep hard" correspond to choosing $q(t) := 1[|t| \geq \alpha]$ to retain large-margin examples (far from the decision boundary) and $q(t) := 1[|t| \leq \alpha]$ to retain small-margin examples (close to the decision boundary), respectively. The notion of an example's difficulty is thus determined by the oracle $w_o$, and the threshold $\alpha > 0$ controls the proportion of data kept. This captures the setting considered in (Sorscher et al., 2022).

**Label-aware Curation.** We also analyze a more realistic rule where the oracle filters for the correctness of the corresponding label as well. Here, an example $(x_i, y_i)$ is kept if its label $y_i$ matches the oracle's label $y_i^o$ and it is deemed interesting by $q$:

$$p_i = 1 \quad \text{iff} \quad y_i = y_i^o \quad \text{and} \quad q(x_i^\top w_o) = 1, \tag{6}$$

where $y_i^o := \text{sign}(x_i^\top w_o)$ is the label according to the pruning oracle (not revealed to the learner!).

In the practical setting of LIMO (Ye et al., 2025) and s1 (Muennighoff et al., 2025) methods, the pruning function $q$ might capture other heuristic rules which decides if an example is sufficiently diverse or interesting to be retained in the curated dataset.

> **Desiderata:** Importantly, our setup posits that the machine learner can only query the curation rule by submitting input/label pairs $(x_i, y_i)$ and obtaining bits $p_i \in \{0, 1\}$, but has no access to the underlying pruning direction $w_o$, nor the oracle labels $y_i^o = \text{sign}(x_i^\top w_o)$.

**Remark 1.** *The setups in Feng et al. (2025) and Firdoussi et al. (2024) are a special case of Eqn (6). This occurs when the difficulty-based pruning is ignored ($q \equiv 1$), meaning the curation rule retains an example if and only if its label $y_i$ matches the oracle's label $y_i^o$.*

**Pruning Ratio.** The fraction of data retained for learning is the **pruning ratio**, $p := \mathbb{P}(p_i = 1)$. Out of $n$ original examples, approximately $np$ survive curation. A small $p$ corresponds to aggressive pruning, while $p \to 1$ means no data is removed.

## 2.3 QUANTIFYING GENERATOR AND PRUNING ORACLE QUALITY

The following constants play a crucial role in our theory. They measure the geometric alignment between the generator (the labeler of the training data, $w_g$), the oracle (the pruner, $w_o$), and the ground truth (the true labeler of the test data, $w_*$):

$$\rho := \frac{w_g^\top C w_*}{\|w_g\|_C \|w_*\|_C}, \quad \rho_* := \frac{w_o^\top C w_*}{\|w_o\|_C \|w_*\|_C}, \quad \rho_g := \frac{w_o^\top C w_g}{\|w_o\|_C \|w_g\|_C}, \quad \tau := \frac{\rho_g}{\sqrt{1 - \rho_g^2}}. \tag{7}$$

Here, $\|w\|_C := \sqrt{w^\top C w}$ is the Mahalanobis norm induced by the covariance matrix $C$.

Geometrically, $\rho$, $\rho_*$, and $\rho_g$ are the **cosines of the angles** between their respective vector pairs, while $\tau$ is the **cotangent** of the angle between the pruner ($w_o$) and the generator ($w_g$).

Crucially, $\rho$ and $\rho_*$ directly quantify the performance of the generator and the pruner. Their test errors are given by the simple relationship:

$$E_{\text{test}}(w_g) = (1/\pi) \arccos \rho \quad \text{and} \quad E_{\text{test}}(w_o) = (1/\pi) \arccos \rho_*.$$

Note that $\arccos$ has range $[0, \pi]$. These constants have the following interpretation for our analysis:

- **Generator Quality ($\rho$):** When $\rho \to 1$, the generator is excellent, which we call a **strong generator**. When $\rho < 1$ corresponding to label shift, it is a **weak generator**.
- **Oracle Quality ($\rho_*$):** When $\rho_* \to 1$, the pruning oracle is excellent and aligns well with the ground truth.

The triplet $(\rho, \rho_g, \rho_*)$ will appear in our analytical descriptions of the limiting test error $E_{\text{test}}(\hat{w})$.

## 3 MAIN THEORY: WHEN TO PRUNE AND WHEN TO SCALE

We established a precise mathematical framework in Section 2, defining key quantities such as the data distribution, model, and curation rules. In this section, we use this framework to develop a core theory that explains when and why data pruning can improve performance by deriving exact scaling laws for test error under different data curation strategies. As we will demonstrate, our theory shows precisely how the optimal pruning strategy changes as a function of $\rho$.

For simplicity, we present our main results for the isotropic setting where $\Sigma = C_g = I_d$ and the pruning direction $w_o$ has unit norm. General results are in the appendix.

**Assumption 1** (Symmetric Pruning Functions). *$q$ is a symmetric binary-valued measurable function, i.e., $q(t) = q(-t) \in \{0, 1\}$ for all $t \in \mathbb{R}$. $\mathcal{Q}$ denotes the collection of all such functions.*

This is a common setup that includes rules based on the absolute value of margins, such as keeping the "easiest" or "hardest" examples (Sorscher et al., 2022).

### 3.1 SETTING #1: LABEL-AGNOSTIC DATA CURATION

We first consider label-agnostic pruning, where the decision to keep an example $(x_i, y_i)$ depends only on the features $x_i$, as in Eqn (5). For any pruning function $q \in \mathcal{Q}$, we define four key constants that capture its effect on the learning dynamics:

$$p := \mathbb{E}\left[q(G)\right], \quad \gamma := \mathbb{E}\left[q(G)G^2\right], \quad \beta := 2\mathbb{E}\left[q(G)\varphi(\tau G)\right], \quad \tilde{\beta} := 2\mathbb{E}\left[q(G)\Phi\left(\tau G\right)G\right], \quad (8)$$

where $\varphi$ and $\Phi$ are the PDF and CDF respectively of a standard Gaussian variable $G \sim \mathcal{N}(0, 1)$. Note that $p = p(q)$ defined above is just the average fraction of data kept by the pruning strategy in Eqn (5).

The following theorem provides our first main result: an exact analytical formula for the test error.

**Theorem 1** (Exact Test Error). *In the asymptotic limit Eqn (4), the test error of the model $\hat{w}$ from Eqn (3) is given by,*

$$E_{test}(\hat{w}) \to \frac{1}{\pi} \arccos(|m_0|/\sqrt{\nu_0}), \text{ where} \quad (9)$$

$$m_0 := \omega m(-\lambda) + \tilde{\omega}\tilde{m}(-\lambda), \quad \nu_0 := p\phi m'(-\lambda) + r'(-\lambda) - \frac{2\phi m'(-\lambda)r(-\lambda)}{1 + \phi m(-\lambda)}, \quad (10)$$

$$\text{with } \omega := (\rho - \rho_g \rho_*), \quad \tilde{\omega} := \tilde{\beta}\rho_*, \quad (11)$$

*where $m$, $\tilde{m}$, and $r$ are functions explicitly determined by the constants in Eqn (8). In particular, $m$ is the Stieltjes transform of a Marchenko-Pastur law, "deformed" by pruning. Details in appendix.*

This theorem provides the machinery to analyze any pruning strategy $q$, and isolate its effect on the dynamics of the classification test error curve. This impact is entirely captured by the scalars $p, \gamma, \beta$, and $\tilde{\beta}$. Now, we use this tool to characterize the optimal choice of $q$.

**Sketch of Proof of Theorem 1.** The full proof is given in the appendix, and relies on the construction of suitable deterministic equivalents for the resolvent matrix $R$ defined in Eqn (3) and its square $R^2$. This allows us to calculate the limiting distribution of the "margin" $yx^\top \hat{w}$ at a random test point $x \sim \mathcal{N}(0, I_d)$, and then the test error $E_{test}(\hat{w}) := \mathbb{P}(yx^\top \hat{w} < 0)$. Our approach follows random matrix theory (RMT) techniques which are now prevalent in machine learning theory (Couillet & Liao, 2022; Firdoussi et al., 2024).

**Optimal Pruning Strategy.** In the asymptotic limit Eqn (4), let $F(q)$ be an error functional representing the limiting test error for a given strategy $q$ in the data-rich, unregularized regime:

$$F(q) := \lim_{\phi \to 0} \lim_{\lambda \to 0} \lim_{d,n \to \infty, \, d/n \to \phi} E_{test}(\hat{w}), \quad (12)$$

where $\hat{w} = \hat{w}(q, n, d, \lambda, \rho_*, \ldots)$ is the estimator Eqn (3) fitted on a version of the training dataset $D_n$ pruned with the pruning strategy $q$.

The following theorem shows how the minimizer of $F(q)$ changes based on the generator quality $\rho$.

**Theorem 2** (Optimal Pruning Strategy). *Suppose that the pruning direction $w_o$ has a positive projection along the generator direction $w_g$ ($\rho_g > 0$) and fix the pruning ratio $p \in (0, 1]$. Let $\mathcal{Q}_p$ be the set of strategies that keep a fraction $p$ of the data.*

*(A) If the generator is excellent ($\rho \to 1$) and the pruner is excellent ($\rho_* \to 1$), then the **"keep hard"** (KH) strategy uniquely minimizes the test error $F(q)$ over $\mathcal{Q}_p$.*

*(B) If the generator is poor ($\rho < 1$) but the pruner is excellent ($\rho_* \to 1$), then the **"keep easy"** (KE) strategy uniquely minimizes the test error $F(q)$ over $\mathcal{Q}_p$.*

Part (A) shows that for a strong model/generator that has already mastered the task, performance is refined by focusing on difficult examples—a "less is more" (Ye et al., 2025) approach. Part (B) captures the opposite scenario: for a weak model/generator, the best strategy is to keep easy examples. This latter case is particularly relevant for mitigating model collapse, where a model trained on its own imperfect outputs acts as a poor generator (Shumailov et al., 2024; Dohmatob et al., 2025). Also see Appendix C.

### 3.2 SETTING #2: LABEL-AWARE DATA CURATION

We now extend our analysis to the pruning rule from Eqn (6), inspired by methods like LIMO (Ye et al., 2025) and s1 (Muennighoff et al., 2025). Here, an example is kept only if an oracle deems its label to be correct **and** it satisfies the difficulty-based rule. This requires modifying the definitions of our key constants from Eqn (8). Set $z_i := x_i^\top w_g$, $z_i^o := x_i^\top w_o$, and $f_i := p_i y_i$, where $p_i \in \{0, 1\}$ is as defined in Eqn (6). The said modifications are:

$$p := \mathbb{P}(p_i = 1), \quad \gamma := \mathbb{E}[(y_i^o)^2 p_i], \quad \beta := \mathbb{E}[\frac{\partial f_i}{\partial z_i}], \quad \tilde{\beta} := \mathbb{E}[\frac{\partial f_i}{\partial z_i^o}]. \tag{13}$$

Expectations are over the training data and derivatives are in the distribution-theoretic sense. Explicit formulae for the above constants are provided in the appendix for a general pruning strategy $q \in \mathcal{Q}$.

**Theorem 3** (Test Error for Label-aware Curation). *In the asymptotic limit Eqn (4), the test error $E_{test}(\hat{w})$ for label-aware curation is given by the same formula as in Theorem 1, but using the modified constants from Eqn (13).*

Refer to the appendix for full proofs, various corollaries and their phenomenological implications.

## 4 BRIDGING THEORY AND PRACTICE

Our theoretical framework provides a clear principle: the optimal data curation strategy is not universal but depends on the interplay between the generator's quality ($\rho$), the pruner's quality ($\rho_*$), their alignment ($\rho_g$), and the amount of available data ($n$). In this section, we first validate our predictions in a controlled synthetic environment. We then use these validated principles as a lens to interpret and unify real-world results in LLM mathematical reasoning and ImageNet classification. For a comprehensive set of validations, please see Figure 4 and Appendix B.

### 4.1 THEORY PREDICTION: THE INTERPLAY OF GENERATOR QUALITY AND DATA SCALE

We simulate four distinct learning regimes in a 2x2 grid to characterize the test error as we vary the generator's quality ($\rho$) and the amount of available data ($n$). The left column shows a **strong generator** ($\rho = 1$), while the right shows a **poor generator** ($\rho < 1$). The top row represents a **small-$n$** regime, and the bottom represents a **large-$n$** regime.

In each setting, we compare a strategic "keep hard" pruning strategy against a baseline "random" selection of the same size, where the pruner is uninformative[1]. Figure 1 plots the test error, showing the match between our theoretical predictions and the empirical results.

The results reveal a clear pattern for when to prune. In three of the four regimes, the test error is minimized when the pruning fraction $p = 1$, confirming the "more is more" (Sun et al., 2025) principle. This holds true when:

---

[1] For the "keep hard" strategy, we set $\rho_g = 0.5$ and $\rho_* = \rho$. The "random" strategy uses an orthogonal pruner where $\rho_* = \rho_g = 0$.

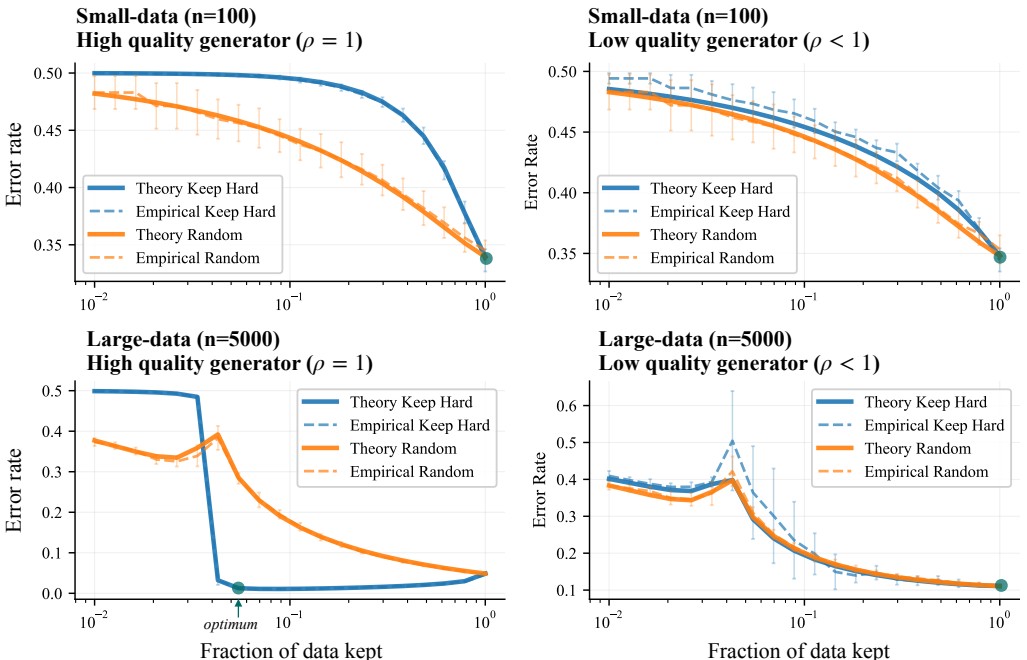

Figure 1: **Theory Prediction across four key regimes.** Test error as a function of fraction of data kept ($p = 1$ means keeping all the data) for "keep hard" and "random" pruning. Solid lines are theoretical predictions; dashed lines are empirical results with error bars. The plot reveals that a "more is more" strategy (optimal error at p=1) is the default, holding true for small datasets (top row) or a poor generator (right column). The bottom-left quadrant shows the crucial exception: only when data is abundant and the generator is strong does the "less is more" principle apply, with aggressive pruning yielding the lowest error.

- The amount of data is small (top row, both poor and strong generators).

- The generator is poor, even with abundant data (bottom right).

However, the bottom-left quadrant reveals the critical exception. When the data is abundant **and** the generator is strong ($\rho = 1$), the error is minimized at $p \ll 1$. This confirms the "less is more" principle: in this specific regime, curating a small set of hard examples is the optimal strategy.

## 4.2 RECONCILING RECENT FINDINGS IN LLM MATH REASONING

Our framework can interpret and unify seemingly contradictory findings in LLM mathematical reasoning. The following results are aggregated from existing literature and our theory provides a novel explanation for *why* different curation strategies succeed under different conditions. In this context, the **generator** ($w_g$) is the base LLM that produces reasoning traces, and its **quality** ($\rho$) reflects its proficiency on a specific slice of the test data.

Recent methods like LIMO and s1 show that "less is more": aggressive curation of high-quality, difficult examples improves *average* performance on the AIME benchmark (Table 1). However, a paradox emerges when evaluating only on the *hardest* AIME questions: here, "more is more" holds true, and performance scales with the number of training examples (Table 2).

Our theory resolves this cleanly:

- **For Average Performance**, the base LLM is a **strong generator** (high $\rho$) for the majority of problems. As predicted by our theory, the optimal strategy is to aggressively prune and "keep hard" examples to refine its already strong capabilities.

Table 1: AIME 2024 (Average Performance) reported in Muennighoff et al. (2025); Ye et al. (2025).

| Training Data Size | Pass@1 (%) |
|---|---|
| 0 (Base Qwen2.5_32B) | 16.5 |
| 114k (Openthinker) | 50.2 |
| 59k (curated in s1) | 53.3 |
| 1k (curated from pool of 59k) | 56.7 |

Table 2: AIME (Hard-Level Questions) performance reported in Sun et al. (2025).

| Training Data Size | Avg@8 (%) |
|---|---|
| 0 (Base Qwen2.5_32B) | 1.0 |
| 1k from OpenR1-Math | 28.4 |
| 2k examples | 35.4 |
| 10k examples | 52.1 |
| 114k (Openthinker) | 47.9 |
| 1M (Openthinker2) | 64.9 |

- **For Hard Performance**, the same LLM is a **weak generator** (low $\rho$) relative to this difficult data slice. In this regime, our theory correctly predicts that a "more is more" approach is superior, as the model needs a larger dataset to build foundational skills for these novel problems.

The optimal strategy is not universal; it depends on the generator's capability relative to the target task's difficulty.

### 4.3 Curation on ImageNet: Data Scale and Model Collapse

We demonstrate that the same principles apply to large-scale vision tasks. We use a pre-trained model as both the *generator* ($w_g$) and *pruner* ($w_o$) to create and select from a pseudo-labeled dataset. The strength of this generator is controlled by the size ($n$) of its initial training set.

**Optimal Strategy Depends on Data Scale.** As predicted, the initial data size dictates the best pruning strategy. Figure 2 shows a clear crossover point:

- **Small $n$ (Weak Generator):** When trained on only 160K examples, the "keep easy" strategy is more effective.

- **Large $n$ (Strong Generator):** When trained on 1.2M examples, the "keep hard" strategy becomes superior, achieving performance close to a model trained on ground-truth labels.

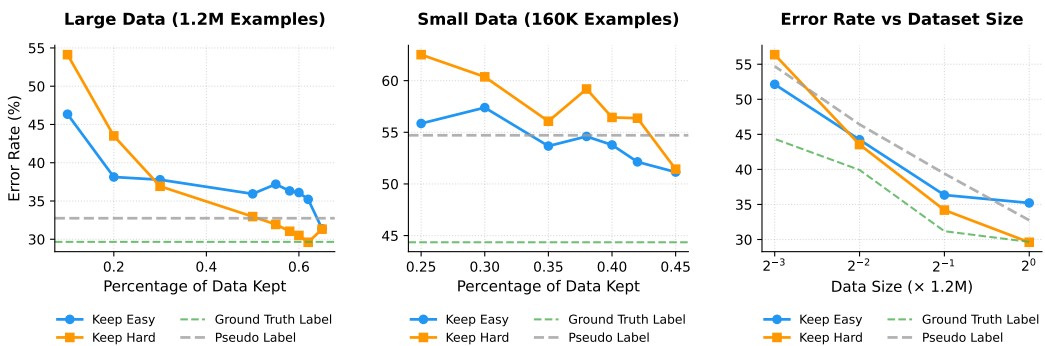

Figure 2: **The optimal curation strategy depends on the data scale in ImageNet.** A clear crossover point emerges as we vary the initial dataset size $n$, shifting the optimal strategy from "keep easy" to "keep hard" as the generator model becomes stronger.

**Strategic Pruning Prevents Model Collapse.** This principle is vital for stability in iterative training. We simulate model collapse by repeatedly re-training on the model's own pseudo-labels. Figure 3 shows that while training on all data causes performance to degrade, applying the "keep hard" strategy at each step stabilizes performance and effectively prevents collapse. This demonstrates that principled curation is crucial not only for one-shot efficiency but also for long-term stability in self-improvement loops.

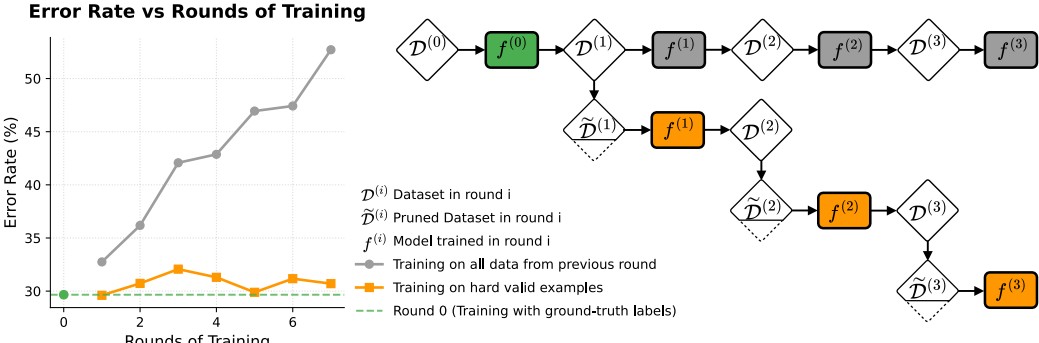

Figure 3: **Strategic pruning prevents model collapse.** Over multiple rounds of pseudo-labeling, training on all examples leads to performance degradation. In contrast, selectively training on only hard, valid examples consistently preserves performance across rounds.

## 5 RELATED WORK

**Beating Neural Scaling Laws.** The award-winning work of Sorscher et al. (2022) show that pruning a training set with margin-based difficulty scores can bend neural scaling curves, delivering higher accuracy with fewer samples. More recent methods in reasoning and program-synthesis tasks—LIMO (Ye et al., 2025) and S1 (Muennighoff et al., 2025) report an even more drastic picture: a compact set of challenging, high-quality examples drives larger gains than indiscriminate data expansion. In these pipelines the *inputs* (questions) are human-curated, while the *outputs* (answers or solutions) are generated by a large model such as R1 (Guo et al., 2025). We provide theoretical justification for the improved scaling behavior and systematically study a simpler, yet analogous, setup through controlled experiments on ImageNet (Deng et al., 2009).

In this line, let us also mention the work of Charton & Kempe (2024) who have studied the effect of repeated training examples for learning mathematical tasks like computing greatest common divisors (GCD) and numerical estimation of matrix eigenvalues, and have reported improved scaling laws.

**Model Collapse.** Advances in generative models have led to synthetic data becoming widespread online, where it now irreversibly blends into training corpora. Recent studies have highlighted the potential for dramatic deterioration in downstream models, a phenomenon known as *"model collapse"* (Shumailov et al., 2023). Empirical studies have demonstrated this issue in various settings (Hataya et al., 2023; Martínez et al., 2023a;b; Bohacek & Farid, 2023; Briesch et al., 2023). Synthetic data can exacerbate biases via feedback loops (Taori & Hashimoto, 2023; Wyllie et al., 2024), narrow content diversity (Padmakumar & He, 2024; Guo et al., 2023), and distort underlying distributions (LeBrun et al., 2021).

Theoretical analysis also examines the effects of iterative training on self-generated data (Alemohammad et al., 2023; Bertrand et al., 2023; Dohmatob et al., 2024a; Seddik et al., 2024). Notably, Dohmatob et al. (2024b) warns that model collapse signifies a break in customary neural scaling laws (Kaplan et al., 2020; Hoffmann et al., 2022), where increasing synthesized data volume does not enhance performance as effectively as scaling with human-generated data. As a result, recent works have focused on avoiding or correcting synthetic data to prevent model collapse. Gillman et al. (2024) propose using a correction function informed by expert knowledge to modify the synthesized data. Alemohammad et al. (2024) leverage a model trained on synthetic data as negative guidance for diffusion models. Zhang et al. (2024) employ the confidence score and an AI detection classifier to discard synthesized data. In contrast, we propose leveraging the synthesized data through strategic selection techniques.

We also note the approach proposed by Gerstgrasser et al. (2024), which suggests accumulating multiple versions of the training dataset over time so that their union, unlike the latest version alone, retains crucial information about the ground truth distribution of the data. While this is an interesting direction, we believe it may face practical limitations as both models and datasets continue to scale over time.

Building on the recent works of Feng et al. (2025); Firdoussi et al. (2024) which assume a pruning oracle that can only guess which examples from the training data have correct labels, we propose and analyze a more general setup covering oracles which can also assess the difficulty of example.

**Benefits of Synthesized Data.** Synthetic data holds great potential, as it is much easier and cheaper to scale compared to human-labeled data. Numerous empirical studies have demonstrated the benefits of synthesized data across a wide range of settings. Common practices include cases where the downstream task slightly differs from that of the data-generating model (Cheng et al., 2024), where the generating model is significantly stronger than the consuming one (Hemmat et al., 2025), or when better prompt engineering and external information are utilized (Shin et al., 2023; Hemmat et al., 2023; Nalela, 2025). Data selection is already employed in some domains, particularly in code generation and mathematics, where natural verifiers such as compilers, solutions, or heuristic verifiers exist. For instance, Haluptzok et al. (2022) generate synthesized code and filter out incorrect samples. Ulmer et al. (2024) use conversational metrics to filter synthetic dialogue data. Trinh et al. (2024) utilize a symbolic deduction engine to verify correct solutions for Olympiad geometry problems. Setlur et al. (2024) apply a final answer verifier to distinguish between good and bad synthetic data. Although verifiers are used in these cases, their effects on performance have not been systematically explored, especially in terms of how different types of verifiers influence outcomes.

## 6 CONCLUDING REMARKS

We put forward a principled view of aggressive data curation, demonstrating that the striking results from systems like LIMO and s1 are not coincidences but follow from fundamental properties of learning with pruned data. By supplying a clean theoretical lens—validated on synthetic data and ImageNet, and shown to explain phenomena in LLMs—we give practitioners a clearer picture of *when* to discard data and *why* this can stabilize training and improve generalization. In doing so, we shift the focus from a "more is always better" mindset toward a more evidence-based, data-centric workflow.

Furthermore, our framework explains how principled curation can mitigate **model collapse** Shumailov et al. (2024), a phenomenon characterized by a shift in scaling laws Dohmatob et al. (2024b;a; 2025). By revealing the stabilizing role of a strong pruning oracle, our findings also provide a theoretical basis for recent empirical successes in this area Feng et al. (2025).

**Limitations.** While our framework provides a unifying perspective, we acknowledge its limitations. Our core theory assumes a high-dimensional Gaussian feature model and binary classification, whereas real-world data is structured, multi-class, and often curated online. We do not address non-linear predictors, the effects of multi-epoch optimization, or the interplay between pruning and active learning.

**Future Directions.** We see three immediate avenues for extending this work:

  (i) *Analysis of non-linear models.* Extending the theory to neural networks in the kernel regime, i.e., random-feature and kernel regimes—or to the infinite-width neural tangent kernel—would bridge the gap to practical deep learning architectures. Such an analysis can still be carried out using RMT ideas. Less obvious is analyzing the feature-learning regime (e.g., SGD on moderately parametrized networks). Here, the analysis becomes significantly more difficult and we can no longer rely on classical RMT. This is an interesting future direction.

 (ii) *Adaptive curation loops.* Incorporating iterative re-scoring and re-training would capture the feedback dynamics used in modern self-distillation and RLHF pipelines.

(iii) *Broader evaluation.* Testing theory-guided pruning on diverse modalities (text, code, speech) and assessing its impact on fairness, privacy, and energy consumption will clarify when and how "less is more" in large-scale ML.

We hope this work provides a rigorous starting point for these efforts and for the principled design of future data-centric training pipelines.

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

# Appendix for "Why Less is More (Sometimes): A Theory of Data Curation"

## CONTENTS

## A  EXPERIMENTAL DETAILS FOR IMAGENET

We now provide details for the experimental results presented in Section 4.3 of the manuscript.

### A.1  DATASET

All experiments are conducted on the ImageNet-1K (Deng et al., 2009) dataset, which contains approximately 1.2 million training images and 50,000 validation images across 1,000 classes. For

experiments with reduced dataset sizes, we use random subsampling to generate smaller training sets at various fractions (e.g., 50%, 25%, 12.5%) of the full dataset.

## A.2 MODEL ARCHITECTURE

We use the Vision Transformer (ViT-B/16) (Dosovitskiy et al., 2020) as our primary backbone, implemented via the MMPretrain framework (Contributors, 2023). The model uses a patch size of 16 and an input resolution of $224 \times 224$. We apply a drop path rate of 0.1 and label smoothing with a smoothing value of 0.1 in the classification head. During training, we apply data augmentation techniques including Mixup ($\alpha = 0.8$) and CutMix ($\alpha = 1.0$).

## A.3 TRAINING SETUP

All models are trained using the AdamW optimizer. The learning rate is scaled with global batch size according to the linear scaling rule. For ViT experiments, the base learning rate is $1 \times 10^{-4} \times \frac{\text{batch size}}{256}$, with a weight decay of 0.3, $\epsilon = 1 \times 10^{-8}$, and $\beta = (0.9, 0.95)$.

To ensure fairness across dataset sizes, we adjust the number of training epochs inversely proportional to the dataset fraction, so that the total number of iterations remains constant.

Training is performed on 4 nodes, each with 8 NVIDIA H100 GPUs (total 32 GPUs), using PyTorch's Distributed Data Parallel (DDP) via SLURM. The batch size per GPU is 128. We use synchronized batch normalization and standard augmentations including random resized crops, horizontal flips, RandAugment, and random erasing. Models are evaluated on the standard ImageNet-1K validation set using top-1 accuracy.

# B EMPIRICAL CONFIRMATION OF OUR THEORETICAL FORMULAE

.

We validated our framework through extensive simulations and comparison with theory, summarized in Figure 4. Synthetic datasets were generated under the model of Section 2, with $d = 200$, varying sample size $n$, pruning fraction $p$, and generator angle $\rho$. Logistic regression with $\lambda = 10^{-6}$ was trained on curated subsets, and error was measured as the angular deviation between learned and true weights.

**Coverage.** We tested 15 parameter settings ($n \in \{500, 1000, 2000\}$, $p \in \{0.2, 0.5, 0.8\}$, $\rho \in \{0, \pi/12, \pi/6, \pi/4\}$, keep-easy vs. keep-hard), spanning both typical and extreme regimes.

**Agreement.** Theoretical and empirical results matched closely: mean relative error $1.8\%$, all $< 5\%$. Bland–Altman analysis showed mean difference 0.0019 with 95% limits of agreement $[-0.0039, 0.0077]$.

**Sweeps and Landscapes.** Parameter sweeps confirmed that theory captures observed non-monotonic pruning effects, power-law scaling with $n$, and angular dependence. Two-dimensional landscapes (sample size $\times$ pruning fraction) showed near-identical patterns, with maximum absolute differences $< 0.01$.

**Statistical Checks.** Empirical error distributions (50 runs) centered tightly around theoretical predictions, and theory lay within 95% confidence intervals across all tested settings.

**Robustness.** Agreement held across configurations, including edge cases ($\rho = 0$, extreme pruning), indicating the framework captures the essential mechanisms.

**Implication.** These results establish that our theory accurately predicts generalization under pruning in high-dimensional linear classification, providing a reliable tool for analyzing and optimizing data curation strategies.

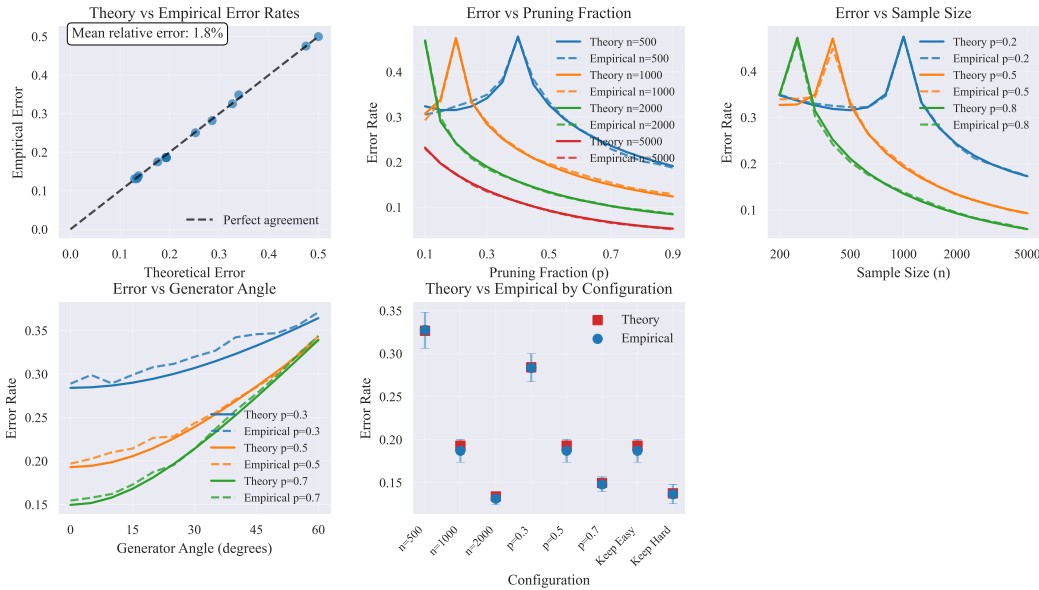

Figure 4: **Validation of theoretical error predictions against empirical simulations.** (A) Scatter plot of theory vs. empirical error across 15 configurations, with diagonal = perfect agreement. (B–D) Parameter sweeps for pruning fraction, sample size, and generator angle. (E) Configuration-wise comparisons. All results use logistic regression with $\lambda = 10^{-6}$.

### B.1  EXPERIMENTS FOR LABEL-AGNOSTIC CURATION RULE EQN (5)

As promised in the main manuscript, Figure 5 presents results on toy data, with curation done according to the label-agnostic rule Eqn (5).

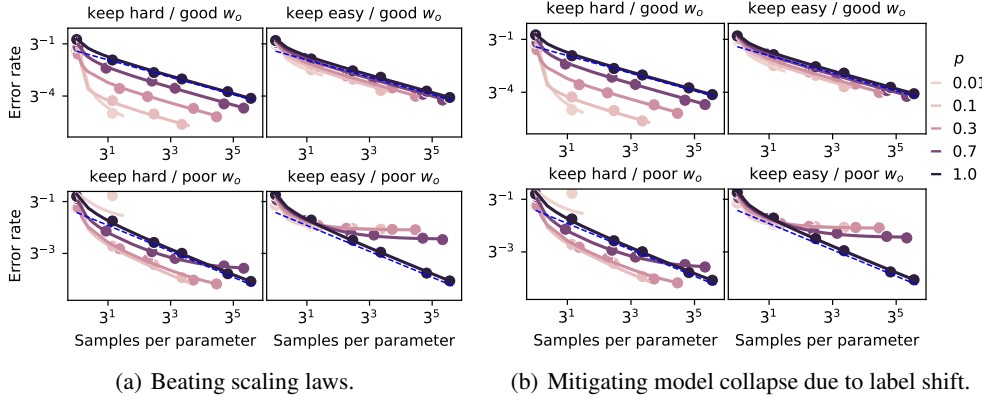

    (a) Beating scaling laws.            (b) Mitigating model collapse due to label shift.

Figure 5: Effect of Label-agnostic curation rule Eqn (5) as proposed in (Sorscher et al., 2022).

### B.2  WHICH IS BETTER, "KEEP EASY EXAMPLES" OF "KEEP HARD EXAMPLES"?

See Figures 6 and 7.

The data is Gaussian, generated according to Eqn (1) with $C = I_d$ (covariance matrix of samples, under the generators distribution) and $\Sigma = I_d$ (ground-truth covariance matrix). The sample size $n$ sweeps the range 10 through $10^6$ in log-scale, while the input dimension fixed to $d = 200$. The data curation is done according to the Label-aware rule Eqn (6). The estimator $\hat{w}$ defined in Eqn (3) is

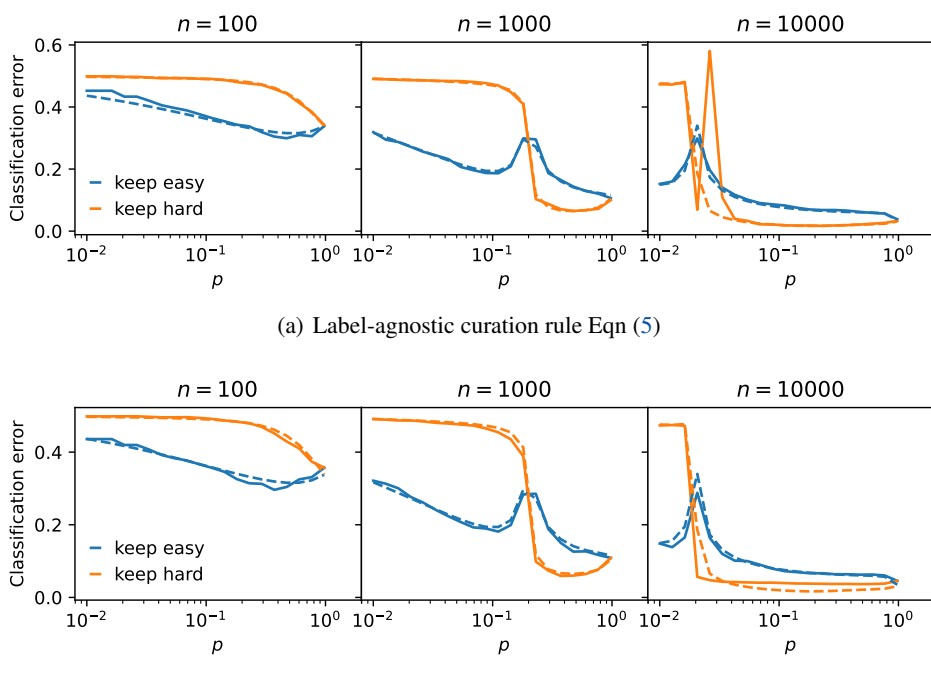

(a) Label-agnostic curation rule Eqn (5)

(b) Label-aware curation rule Eqn (6)

Figure 6: **Beating scaling laws.** Solid lines are experiments; broken lines are our theoretical predictions (Theorem 1 and Theorem 3). For this experiment, the angle between generator labeling vector $w_g$ is perfect, i.e $w_g = w_*$, the ground-truth. Notice the perfect agreement between theory and experiment.

computed using Scipy's linear algebra functions operations (from the "linalg" module therein), with regularization parameter fixed at $\lambda = 10^{-6}$. The classification test error $E_{test}$ is defined as:

$$E_{test}(\hat{w}) := \mathbb{E}\left[\ell_{0/1}(\text{sign}(x^\top \hat{w}), y)\right] = \mathbb{P}(\text{sign}(x^\top \hat{w}) \neq y). \tag{14}$$

.

The pruning direction $w_o$ in Eqn (6) is chosen to make an angle $\theta = 0$ (perfect pruning direction) or $\theta = \pi/10$ (poor pruning direction) with the ground-truth labeling vector $w_* = (1, 0, \ldots, 0)$.

For Figure 5(a) ("beating neural scaling laws"), the labeling vector $w_g \in \mathbb{R}^d$ for the generator equals that of the ground-truth. Thus, the generator is taken to be perfect, a setting also considered in (Sorscher et al., 2022).

For Figure 5(b) ("mitigating model collapse"), the generator is imperfect: its labeling vector $w_g$ makes an angle $\pi/5$ with the ground-truth $w_*$. This imperfection simulates the model collapse phenomenon (Shumailov et al., 2024; Dohmatob et al., 2024a;b; Feng et al., 2025; Dohmatob et al., 2025).

## C RESULTS IN THE REGRESSION SETTING

### C.1 THEORETICAL SETUP

As promised in the main paper, we now turn to the case of regression, where the label variable $y$ in the data distribution Eqn (1) is now given by

$$y = x^\top w_* + \eta, \tag{15}$$

where $\eta \sim \mathcal{N}(0, \sigma^2)$ is a noise variable independent of the covariates $x$. The test error of the estimator $\hat{w}$ is now measured by

$$E_{reg}(\hat{w}) := \mathbb{E}_{(x,y)\sim P_*}[(x^\top \hat{w} - x^\top w_*)^2] - \sigma^2. \tag{16}$$

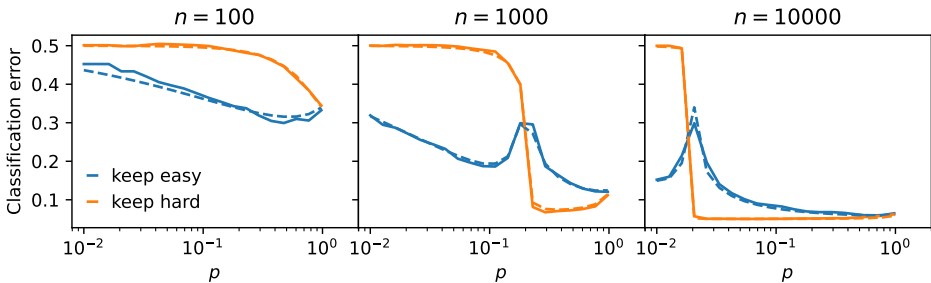

(a) Label-agnostic curation rule Eqn (5) (proposed in (Sorscher et al., 2022))

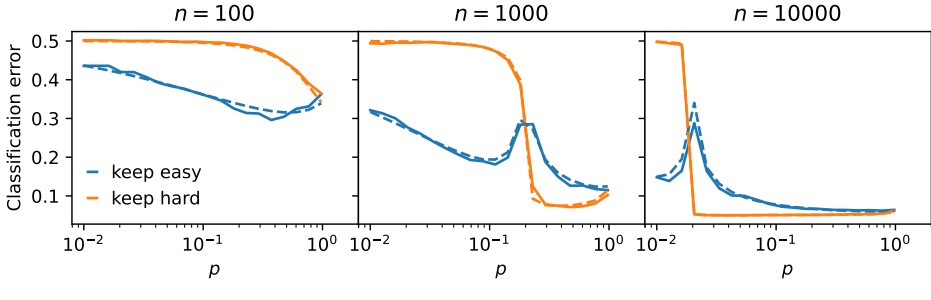

(b) Label-aware curation rule Eqn (6)

Figure 7: **Mitigating model collapse.** Solid lines are experiments; broken lines are our theoretical predictions (Theorem 1 and Theorem 3). For this experiment, the angle between generator labeling vector $w_g$ and ground-truth $w_*$ is $\pi/20$, thus simulating an imperfect generator. Notice the perfect agreement between theory and experiment.

## C.2 MAIN RESULT FOR REGRESSION

Define the following auxiliary quantities

$$w_g^{\not/} := (w_g^\top w_o)w_o, \; w_g^\perp := w_g - w_g^{\not/}, \; \epsilon := w_g - w_*, \; a := \epsilon^\top w_g^\perp, \; b := \epsilon^\top w_g^{\not/}, \; c^2 := \|\epsilon\|^2. \quad (17)$$

Thus, $w_g^{\not/}$ is the component of $w_g$ pointing in the direction of the pruning vector $w_o$ and $w_g$ is the perpendicular component. $c^2$ measures the disparity between the generative and the ground-truth labeling vectors $w_g$ and $w_*$ respectively. It is clear that

$$\|w_g^{\not/}\|^2 = \rho_g^2\|w_g\|^2, \quad \|w_g^\perp\|^2 = (1-\rho_g^2)\|w_g\|^2, \quad (18)$$

$$a = \|w^\perp\|^2 - \|w_g\|(\rho - \rho_g\rho_*), \quad b = \|w^{\not/}\|^2 - \|w_g\|\rho_g\rho_*, \quad (19)$$

where $\rho$, $\rho_g$, and $\rho_\star$ are as defined in Eqn (8).

The following is one of our main contributions.

**Theorem 4.** *In the limit Eqn (4), the regression test error of the model $\hat{w}$ defined in Eqn (3) is given by*

$$E_{reg}(\hat{w}) \to B + V + c^2 - 2\lambda \cdot (m(-\lambda)a + \tilde{m}(-\lambda)b),$$

$$\textit{with } B := \lambda^2 \cdot \left(m'(-\lambda)\|w_g^\perp\|^2 + \tilde{m}'(-\lambda)\|w_g^{\not/}\|^2\right), \quad V := \sigma^2\phi\bar{m}'(-\lambda). \quad (20)$$

**Universality.** Note that for a fixed pruning rate $p \in (0, 1]$ and pruning direction $w_o$, the specific choice of pruning strategy $q \in \mathcal{Q}$ used only enters the picture via $\gamma = \gamma(q)$, defines in Eqn (8). Two pruning strategies with the same value of $\gamma$ induces exactly the same test error dynamics $E_{reg}$ in the high-dimensional limit Eqn (4).

**Unregularized Regime.** We now consider our theory in the limit $\lambda \to 0$, in which case the estimator $\hat{w}$ defined in Eqn Eqn (3) reduces to the least-squares estimate for $w_*$, namely $\hat{w} = X'^\dagger Y'$, where $(X', Y')$ is the pruned training dataset, i.e the nonzero rows of $(DX, DY)$.

**Corollary 1.** *In the limit Eqn (4) then $\lambda \to 0$, it holds that $E_{reg} \to L$, where*

*(A) If $\phi < p$, then $L = \dfrac{\sigma^2 \phi}{p - \phi} + c^2$.*

*(B) If $\phi > p$, then with $c_0 := 1 - p/\phi$ and $c_1 = \gamma/\phi + c_0 = 1 - (p - \gamma)/\phi$, we have*

$$L = \frac{\sigma^2}{\phi - p} + (\|w_g^\perp\|^2 + \|w_g^\sslash\|^2/c_1)c_0 + c^2 - 2(a + b/c_1)c_0.$$

Note that when $p = 1$ (corresponding to no pruning), the above result recovers one of the main results of Dohmatob et al. (2025), namely, their Corollary 1.

The following result is yet another important consequence.

---

**Corollary 2.** *In the noiseless setting $\sigma = 0$, the following hold:*

$$\lim_{\substack{\phi \to 0}} \lim_{\lambda \to 0} \lim_{\substack{d,n \to \infty \\ d/n \to \phi}} E_{reg}(\hat{w}) = \|w_* - w_g\|^2 = c^2 \; \forall p \in (0, 1],$$

$$\lim_{\phi \to 0} \inf_{p \in (0,1]} \lim_{\lambda \to 0} \lim_{\substack{d,n \to \infty \\ d/n \to \phi}} E_{reg}(\hat{w}) = \begin{cases} \|w_* - w_g^\sslash\|^2 < c^2, & \text{if } \|w_* - w_g^\sslash\|^2 < c^2 < \|w_* - w_g^\perp\|^2, \\ c^2, & \text{otherwise} \end{cases}$$

---

Thus, pruning provably mitigates model collapse, under the sufficient condition

$$\|w_* - w_g^\sslash\| < \|w_* - w_g\| < \|w_* - w_g^\perp\|.$$

Note that if $\|w_*\|^2 = 1$ and $\|w_g\|^2 = r^2$, then $c^2 = \|w_* - w_g\|^2 = 1 + r^2 - 2r\rho_g$. Furthermore, if $\rho_* = 1$ (i.e $w_o = w_*$), then $\|w_* - w_g^\sslash\|^2 = \|w_* - \rho_g w_*\|^2 = (1 - \rho_g)^2$.

Keep if $|y_i - x_i^\top w_*|^2$

### C.3 Optimal Pruning in Regression Setting

Consider a sub-collection of parametrized pruning strategies constructed as follows. For any $p, u \in [0, 1]$, define $q_{p,u} \in \mathcal{Q}$ by

$$q_{p,u}(t) := \begin{cases} 0, & \text{if } a(p, u) < |t| \le b(p, u), \\ 1, & \text{otherwise,} \end{cases} \tag{21}$$

$$\text{with } a(p, u) := \Phi^{-1}((1 + (1 - u)p)/2), \quad b(p, u) := \Phi^{-1}(1 - pu/2). \tag{22}$$

Thus, $q_{p,u}$ is the indicator function of the disjoint union of 3 intervals: $[-a(p, u), a(p, u)]$, and two "tails" $(-\infty, -b(p, u))$ and $(b(p, u), \infty)$. Such a pruning strategy selects a mixture of "very easy" training examples (corresponding to neighborhood of 0) and "very hard" examples (corresponding to tails). The parameter $p$ controls the proportion of training data that survives pruning, i.e we have $p(q_{p,u}) = p$, while the parameters $u$ controls the fraction thereof which are "very hard".

**Theorem 5.** *For any pruning strategy $q \in \mathcal{Q}$, there exist $p, u \in [0, 1]$ such that pruning strategy $q_{p,u}$ induces the the same regression test error $E_{reg}(\hat{w})$ for the estimator $\hat{w}$ define in Eqn Eqn (3) as pruning with $q$. In particular, the optimal pruning strategy has the form $q_{p,u}$.*

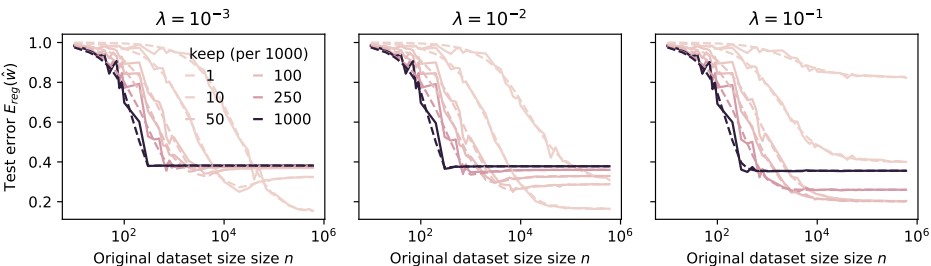

(a) Test error vs original dataset size $n$. We plot the regression test error $E_{reg}$ as a function of the original/unpruned dataset size $d$ and report result for different rates of pruning (per thousand examples). Solid lines correspond to experiments while broken lines correspond to the analytic expression provided by Theorem 4. Notice the perfect match between theoretical predictions and experiment. We see that it is optimal it is optimal consider and unregularized model (small $\lambda$) and discard almost all training data!

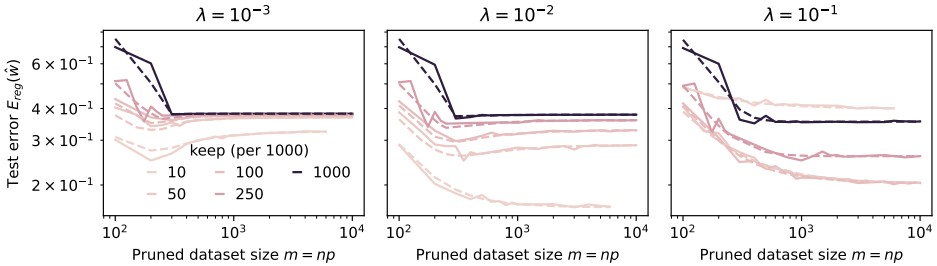

(b) Test error vs pruned dataset size $m = np$. We plot test error as a function of the pruned dataset size $m$ actually used to fit the model, the point being to control for the amount of compute. Once again, we see that it is optimal to discard almost all training data. However, optimal regularization is no longer zero; for nonzero $\lambda$, the error might eventually increase with $m$.

Figure 8: **Mitigating model collapse via pruning** in regression setting. Different colors correspond to different levels of pruning where we keep only the hardest/most informative examples $(x_i, y_i)$ with the largest value of the projection of the features $|x_i^\top w_o|$ along the pruning direction $w_o$.

# D MAIN INGREDIENTS OF PROOFS

## D.1 DETERMINISTIC EQUIVALENT FOR THE RESOLVENT MATRIX $R$

**Definition 1** (Deterministic Equivalents). *Given a sequence of random $N \times N$ matrices $(R_N)_N$, a deterministic equivalent thereof is a sequence of deterministic $N \times N$ matrices $(\overline{R}_N)_N$ such that*

$$\text{tr}\, A_N(R_N - \overline{R}_N) \overset{a.s}{\to} 0, \tag{23}$$

*for all sequences of $N \times N$ matrices $(A_N)_N$ with bounded Frobenious norm.*

Let $\Pi$ (resp. $\Pi^\perp = I_d - \Pi$) be the projection onto the span (resp. orthogonal complement of the span) of $w_o$. Define the following auxiliary vectors and scalars

$$v = \Sigma^{1/2} w_o, \quad v_1 = \frac{v^\top w_o}{\|w_o\|}, \quad v_\perp = \Pi^\perp v. \tag{24}$$

Note that $v_\perp$ is $(d-1)$-dimensional and $\|v_\perp\| = \sqrt{\|v\|^2 - v_1^2}$.

Henceforth we make the replacement $z = -\lambda < 0$, so that the resolvent matrix $R$ now writes

$$R = R(z) := (X^\top D X/n - z I_d)^{-1}, \tag{25}$$

where we recall that $D$ is the $n \times D$ diagonal matrix appearing in Eqn (3), with $D_{ii} = p_i$, the prune/no prune bit for the $i$th training example. Let $\delta(z)$ be the unique positive solution to the fixed-point equation

$$m(z) = d^{-1}\,\text{tr}\,\bar{R}_b(z), \quad \delta(z) = n^{-1}\,\text{tr}\, C\bar{R}_b(z), \quad \bar{R}_b(z) = \left(\mathbb{E}\left[\frac{p_i}{1 + p_i\delta(z)}\right] C - z I_d\right)^{-1}. \tag{26}$$

Note that the inner expectation evaluates to

$$\mathbb{E}\left[\frac{p_i}{1 + p_i\delta(z)}\right] = \frac{p}{1 + \delta(z)} =: t(z),$$

and so $\bar{R}_b(z) = (t(z)C - z I_d)^{-1}$. Observe that $\bar{R}_b(z)(t(z)C - z I_d) = I_d$, and so $t(z)C\bar{R}_b(z) = I_d + z\bar{R}_b(z)$. We deduce that

$$t(z)\delta(z) = n^{-1}\,\text{tr}\, t(z)C\bar{R}_b(z) = n^{-1}\,\text{tr}(I_d + z\bar{R}_b(z)) = \phi \cdot (1 + zm(z)).$$

Thus, the equations defining $m(z)$ and $\delta(z)$ can be rewritten as

$$m(z) = d^{-1}\,\text{tr}(t(z)C - z I_d)^{-1}, \tag{27}$$

$$t(z) = \frac{p}{1 + \delta(z)}, \tag{28}$$

$$\phi \cdot (1 + zm(z)) = t(z)\delta(z) = t(z)\left(\frac{p}{t(z)} - 1\right) = p - t(z). \tag{29}$$

Solving for $\phi zm(z)$ in terms of $t(z)$ in the last equation gives

$$\phi zm(z) = \frac{p\delta(z)}{1 + \delta(z)} - \phi = p - \phi - \frac{p}{1 + \delta(z)} = p - \phi - t(z).$$

Plugging this into the first equation gives the following fixed-point equation for $t(z)$

$$p - \phi - t(z) = zn^{-1}\,\text{tr}(t(z)C - z I_d)^{-1}. \tag{30}$$

The following result shows that $\bar{R}$ is a deterministic equivalent for $R$.

**Proposition 1.** *Recall the function $t(z)$ as the unique positive solution to the equation Eqn (30). Then,*

$$R \simeq \bar{R}, \text{ with } \bar{R} = C^{-1/2}(\check{m}(z)\Pi^\perp + \tilde{m}(z)\Pi)C^{-1/2}, \tag{31}$$

$$\text{where } \check{m}(z) := \frac{1}{t(z) - z}, \quad \tilde{m}(z) := \frac{1}{s(z) - z}, \quad s(z) := \frac{\gamma}{p}t(z). \tag{32}$$

### D.2 THE ISOTROPIC CASE

Consider the special case where the covariance matrix is $C = I_d$. Fix an L2-regularization parameter $\lambda > 0$ and pruning rate $p \in [0, 1]$.

**Lemma 1.** *For every $z = -\lambda < 0$, $m(z)$ is the unique positive solution to the fixed-point equation Eqn (34), and is given explicitly by formula*

$$m(z) = \frac{p - \phi - z - \sqrt{(p - \phi - z)^2 - 4\phi z}}{2\phi z}. \tag{33}$$

Alternatively, $m(z)$ defined in Eqn (33) unique positive solution to the fixed-point equation:

$$\frac{1}{m} = -z + \frac{p}{1 + \phi m}, \text{ with } z := -\lambda. \tag{34}$$

Thus Lemma 1 shows that $m(z)$ is the Stieltjes transform of the limiting spectral density of the resolvent matrix $R$ appearing in Eqn (3), and has the property (among many others) that $d^{-1} \operatorname{tr} R \to m(z)$ in the limit Eqn (4). It represents a somewhat distorted Marchenko-Pastur law; indeed, the classical MP corresponds to $p \to 1$ (i.e. no pruning).

Furthermore, it is not hard to see that

$$\bar{m}(z) \equiv m(z) \equiv \delta(z)/\phi \tag{35}$$

in this case.

*Proof of Lemma 1.* Indeed, observe that in the isotropic case the equation Eqn (30) reduces to $p - \phi - t(z) = \phi z/(t(z) - z)$, or equivalently

$$0 = \phi z + (t(z) - p + \phi)(t(z) - z) = t(z)^2 - (p - \phi + z)t(z) + pz.$$

The discriminant of this quadratic equation evaluates to

$$\begin{aligned}
(p - \phi + z)^2 - 4pz &= (p - \phi - z + 2z)^2 - 4pz \\
&= (p - \phi - z)^2 + 4z^2 + 4z(p - \phi - z) - 4pz \\
&= (p - \phi - z)^2 - 4\phi z,
\end{aligned}$$

and so because $z = -\lambda < 0$, the positive solution is

$$t(z) = \frac{p - \phi + z + \sqrt{(p - \phi - z)^2 - 4\phi z}}{2}. \tag{36}$$

We deduce that

$$\begin{aligned}
m(z) = \frac{1}{t(z) - z} &= \left( \frac{p - \phi - z + \sqrt{(p - \phi - z)^2 - 4\phi z}}{2} \right)^{-1} \\
&= 2 \cdot \frac{p - \phi - z - \sqrt{(p - \phi - z)^2 - 4\phi z}}{(p - \phi - z) - ((p - \phi - z)^2 - 4\phi z)} \\
&= \frac{p - \phi - z - \sqrt{(p - \phi - z)^2 - 4\phi z}}{2\phi z},
\end{aligned}$$

which is precisely the formula given in Eqn (34). □

**Spectral Functions.** Define the following auxiliary functions:

$$\bar{m}(z) := zm(z), \quad s(z) := \frac{\gamma}{1 + \phi m(z)}, \quad \tilde{m}(z) := \frac{1}{s(z) - z}, \quad r(z) := \beta^2 m(z) + \tilde{\beta}^2 \tilde{m}(z), \tag{37}$$

where the constants $\tilde{\beta}$ and $\beta$ are as defined in Eqn (8). Notice that $r$ is (proportional to) a convex combination of $m$ and $\tilde{m}$.

We will be needing the derivatives of $m'$, $\bar{m}'$, $\tilde{m}'$, and $r'$. This is the purpose of the next lemma.

**Lemma 2.** *We have the following identities:*

$$m'(z) = \frac{m(z)^2}{1 - (1 + \bar{m}(z))^2 \phi/p}, \quad \bar{m}'(z) = \frac{p}{(z + \phi\bar{m}(z))^2/\bar{m}(z)^2 - p\phi} = \frac{p}{(\phi + 1/m(z))^2 - p\phi},$$

$$\tilde{m}'(z) = \tilde{m}(z)^2 \left( \frac{\gamma\phi m'(z)}{(1 + \phi m(z))^2} + 1 \right), \quad r'(z) = \beta^2 m'(z) + \tilde{\beta}^2 \tilde{m}'(z).$$

The following result then follows directly from Proposition 1.

**Corollary 3.** *In the isotropic setting, we have the following deterministic equivalents:*

$$R \simeq \bar{R}, \text{ with } \bar{R} = m(z)\Pi^\perp + s(z)\Pi, \tag{38}$$

$$R^2 \simeq m'(z)\Pi^\perp + \tilde{m}'(z)\Pi. \tag{39}$$

*where $\tilde{m}(z) := 1/(s(z) - z)$, $s(z) = \gamma/(1 + \phi m(z))$, and $\gamma \geq 0$ is as given in Eqn (8).*

## D.3 TEST ERROR REPRESENTATION: THE CLASSIFICATION SETTING

WLOG, suppose henceforth that $\bar{w}_g := C^{1/2}w_g$, $\bar{w}_o := C^{1/2}w_o$, and $\bar{w}_* := C^{1/2}w_*$ are unit vectors in $\mathbb{R}^d$. Let $u = \bar{w}_o$ and let $v$ be its completion to an orthonormal basis for the span of $\bar{w}_o$ and $\bar{w}_g$ (if $\bar{w}_o$ and $\bar{w}_g$ are parallel, i.e if $\rho_g = \pm 1$, we simply set $v = 0$). Define $c \in \mathbb{R}^d$ by

$$c := \mathbb{E}[p_i y_i x_i], \tag{40}$$

for a random training data point $(x_i, y_i) \sim P_g$ and corresponding selection/no select bit $p_i \in \{0, 1\}$ (e.g, $p_i$ is as given in Eqn (5) in the case of label-agnostic data curation and Eqn (6) in the case of Label-aware data pruning).

Also define $p = p(q) \in [0, 1]$ and $\gamma = \gamma(q) \geq 0$ by

$$p = \mathbb{E}[p_i], \quad \gamma := \mathbb{E}[(x_i^\top w_o)^2 p_i]. \tag{41}$$

**Lemma 3.** *It holds that $c = \beta_1 C^{1/2}u + \beta_2 C^{1/2}v$, with the $\beta_k$'s as given in Table 3. Also, the constants $p$ and $\gamma$ defined in Eqn (41) are as given in the table.*

| Curation | $p(q)$ | $\gamma(q)$ | $\beta_2(q)$ | $\beta_1(q)$ |
|---|---|---|---|---|
| Label-agnostic | $\mathbb{E}[q(G)]$ | $\mathbb{E}[q(G)G^2]$ | $2\mathbb{E}[q(G)\varphi(\tau G)]$ | $2\mathbb{E}[q(G)\Phi(\tau G)G]$ |
| Label-aware | $\mathbb{E}[q(G)\Phi(\tau|G|)]$ | $\mathbb{E}[q(G)\Phi(\tau|G|)G^2]$ | $\mathbb{E}[q(G)\varphi(\tau G)]$ | $\mathbb{E}[q(G)\Phi(\tau|G|)|G|]$ |

Table 3: **Fundamental constants.** Here, $q \in \mathcal{Q}$ is any even/symmetric pruning function and $G \sim \mathcal{N}(0, 1)$, with pdf $\varphi$ and cdf $\Phi$. Recall that $\tau := \rho_g/\sqrt{1 - \rho_g^2}$, and we use the identification $\beta \to \beta_2$, $\tilde{\beta} \to \beta_1$. Note that taking $q \equiv 1$ on the second row corresponds to the setup of Feng et al. (2025) and Firdoussi et al. (2024).

We are now ready to state our main results, which is a generalization of Theorem 1 and 3.

**Proposition 2.** *Let $c \in \mathbb{R}^d$ be as defined in Eqn (40). For a random test point $(x, y) \sim P_*$, we have the following high-dimensional representation (where $G_1$ and $G_2$ are iid from $\mathcal{N}(0, 1)$):*

$$yx^\top \hat{w} \xrightarrow{L} m|G_1| + \sqrt{\nu - m^2}G_2, \text{ with} \tag{42}$$

$$m \simeq \frac{m_0}{1 + \delta}, \quad m_0 := \frac{c^\top \bar{R}\Sigma w_*}{\|\Sigma^{1/2}w_*\|}, \tag{43}$$

$$\nu \simeq \frac{\nu_0}{(1 + \delta)^2}, \quad \nu_0 := \frac{p}{n} \operatorname{tr}\Sigma C' + c^\top\Sigma'c - \frac{2c^\top \bar{R}c}{1 + \delta}\frac{1}{n}\operatorname{tr}\Sigma C', \tag{44}$$

$$\bar{R} := \mathbb{E}[R], \quad C' := \mathbb{E}[RCR], \quad \Sigma' := \mathbb{E}[R\Sigma R], \tag{45}$$

where $\delta = \delta(-\lambda) > 0$ is as defined by the fixed-point equations Eqn (26).

Furthermore, it holds that

$$E_{test}(\hat{w}) := \mathbb{P}(yx^\top \hat{w} \le 0) \to \frac{1}{\pi} \arccos(|m_0|/\sqrt{\nu_0}). \tag{46}$$

**Remark 2.** *Note that the above result is valid for any curation strategy which maps easy training example $(x_i, y_i)$ to a prune/no prune bit $p_i \in \{0, 1\}$, in an iid fashion. The choices Eqn (5) (label-agnostic) and Eqn (6) (Label-aware) are but particular cases.*

## E    PROOF OF PROPOSITION 2

For a random test point $(x, y) \sim P_*$, we can write

$$yx^\top \hat{w} = yz^\top \Sigma^{1/2} \hat{w} = sign(z^\top \Sigma^{1/2} w_*) z^\top \Sigma^{1/2} \hat{w}.$$

Write $\Sigma^{1/2} \hat{w} = \alpha \Sigma^{1/2} w_* + r$, where $r = \Sigma^{1/2} \hat{w} - \alpha \Sigma^{1/2} w_*$ and $\alpha \ge 0$ is to be determined. Observe that $r$ is perpendicular to $\Sigma^{1/2} w_*$ iff $r^\top \Sigma^{1/2} w_* = \hat{w}^\top \Sigma w_* - \alpha \|\Sigma^{1/2} w_*\|^2 = 0$ iff

$$\alpha = \hat{w}^\top \Sigma w_* / \|\Sigma^{1/2} w_*\|^2. \tag{47}$$

With this choice of $\alpha$, one computes

$$yx^\top \hat{w} = \alpha yz^\top \Sigma^{1/2} w_* + yz^\top r. \tag{48}$$

Because $r$ is perpendicular to $\Sigma^{1/2} w_*$, we know that the above is a sum of two independent random variables.

For the first summand in Eqn (48), observe that

$$yz^\top \Sigma^{1/2} w_* = yx^\top w_* = sign(x^\top w_*) x^\top w_* = |x^\top w_*|,$$

which has the same distribution as $|G|$ for $G \sim N(0, w_*^\top \Sigma w_*)$.

For the second summand, it has distribution $\mathcal{N}(0, \|r\|^2)$ with $\|r\|^2 = \|\Sigma^{1/2} \hat{w}\|^2 - \alpha^2 \|\Sigma^{1/2} w_*\|^2$.

### E.1    ASYMPTOTICS OF $\|\Sigma^{1/2} \hat{w}\|^2$

Now, one computes

$$\hat{w} = \frac{1}{n} \sum_i p_i y_i R x_i = \frac{1}{(1+\delta)n} \sum_i p_i y_i R_{-i} x_i.$$

We deduce that

$$(1+\delta)^2 n^2 \|\Sigma^{1/2} \hat{w}\|^2 = n \sum_i p_i x_i^\top R_{-i} \Sigma R_{-i} x_i + \sum_{i,j,\ j \ne i} p_i p_j y_i y_j x_i^\top R_{-i} \Sigma R_{-j} x_j.$$

Now, observe that

$$\begin{aligned}
\frac{1}{n^2} \sum_i p_i x_i^\top R_{-i} \Sigma R_{-i} x_i &= \frac{1}{n^2} \sum_i \mathrm{tr}(p_i x_i x_i^\top R_{-i} \Sigma R_{-i}) \\
&\simeq \frac{1}{n^2} \sum_i \mathrm{tr}(\mathbb{E}[p_i x_i x_i^\top R_{-i} \Sigma R_{-i}]) \\
&= \frac{p}{n} \mathrm{tr}\, C R_{-i} \Sigma R_{-i} \\
&\simeq \frac{p}{n} \mathrm{tr}\, \Sigma C'.
\end{aligned}$$

For two distinct sample indices $i, j \in [n]$, we have

$$R_{-i} = R_{-ij} - \frac{1/n}{1+\delta} R_{-ij} x_j x_j^\top R_{-ij},$$

$$R_{-i} \Sigma R_{-j} = (R_{-ij} - \frac{1/n}{1+\delta} R_{-ij} x_j x_j^\top R_{-ij}) \Sigma (R_{-ij} - \frac{1/n}{1+\delta} R_{-ij} x_i x_i^\top R_{-ij})$$

$$= R_{-ij} \Sigma R_{-ij} - \frac{1/n}{1+\delta} R_{-ij} \Sigma R_{-ij} x_i x_i^\top R_{-ij} - \frac{1/n}{1+\delta} R_{-ij} x_j x_j^\top R_{-ij} \Sigma R_{-ij}$$

$$+ \frac{1/n^2}{(1+\delta)^2} R_{-ij} x_j x_j^\top R_{-ij} \Sigma R_{-ij} x_i x_i^\top R_{-ij}$$

and so

$$\mathbb{E}[p_i p_j y_i y_j x_i^\top R_{-i} \Sigma R_{-j} x_j] = A_1 - A_2 - A_3 + A_4, \text{ where}$$

$$A_1 := \mathbb{E}[p_i p_j y_i y_j x_i^\top R_{-ij} \Sigma R_{-ij} x_j],$$

$$A_2 := \frac{1/n}{1+\delta} \mathbb{E}[p_i p_j y_i y_j x_i^\top R_{-ij} \Sigma R_{-ij} x_i x_i^\top R_{-ij} x_j],$$

$$A_3 = \frac{1/n}{1+\delta} \mathbb{E}[p_i p_j y_i y_j x_i^\top R_{-ij} x_j x_j^\top R_{-ij} \Sigma R_{-ij} x_j],$$

$$A_4 = \frac{1/n^2}{(1+\delta)^2} \mathbb{E}[p_i p_j y_i y_j x_i^\top R_{-ij} x_j x_j^\top R_{-ij} \Sigma R_{-ij} x_i x_i^\top R_{-ij}]$$

By symmetry, it is clear that $A_4 = 0$. In order to compute $A_2$ and $A_3$, we shall need the following result which can be obtained by applying Wick's identity (aka Anderson-Isserlis arguments).

**Lemma 4.** *Let $x$ and $z$ be iid $\mathcal{N}(0, C)$ and let $g : \mathbb{R}^d \to \mathbb{R}$ be an odd function. Define $c := \mathbb{E}[g(x)x]$. Then, for possibly random random $d \times d$ matrices $A$ and $B$ independent of $x$ and $z$,*

$$\mathbb{E}[g(x)g(z)x^\top A z \mid A] = c^\top A c,$$

$$\mathbb{E}[g(x)g(z)(x^\top A z)(x^\top B x) \mid A, B] = \operatorname{tr}(BC)c^\top A c + 2c^\top A C B c,$$

$$\mathbb{E}[g(x)g(z)(x^\top A z)(x^\top B z)^2 \mid A, B] = \operatorname{tr}(BC)^2 c^\top A c + 4 \operatorname{tr}(BC) c^\top A C B c + 2c^\top A C B C B c.$$

Applying the first part of the lemma with $A = R\Sigma R$ gives $A_1 \simeq c^\top \Sigma' c$, where $\Sigma' := \mathbb{E}[R\Sigma R]$. Applying the second part of the lemma with $A = R_{-ij} \simeq R$ and $B = R_{-ij} \Sigma R_{-ij} \simeq R\Sigma R$ gives

$$A_3 = A_2 \simeq \frac{1}{1+\delta} \frac{1}{n} \left( \operatorname{tr}(\Sigma C') c^\top R c + 2 c^\top R C R \Sigma R c \right)$$

$$\simeq \frac{1}{1+\delta} \frac{1}{n} \operatorname{tr}(\Sigma C') c^\top R c \simeq \frac{c^\top \bar{R} c}{1+\delta} \frac{1}{n} \operatorname{tr} \Sigma C'.$$

We deduce that

$$\|\Sigma^{1/2} \hat{w}\|^2 \simeq \frac{1}{(1+\delta)^2} \left( \frac{p}{n} \operatorname{tr} \Sigma C' + c^\top \Sigma' c - \frac{2c^\top \bar{R} c}{1+\delta} \frac{1}{n} \operatorname{tr} \Sigma C' \right) =: \nu. \tag{49}$$

### E.2 Asymptotics of $\alpha$

**Mean.** One computes

$$\|\Sigma^{1/2} w_*\|^2 \mathbb{E}\alpha = \mathbb{E}\hat{w}^\top \Sigma w_* \simeq \frac{1}{1+\delta} \mathbb{E} \frac{1}{n} \sum_i p_i y_i x_i^\top R_{-i} \Sigma w_*$$

$$\simeq \frac{1}{1+\delta} \mathbb{E}[p_i y_i x_i^\top R_{-i} \Sigma w_*]$$

$$= \frac{1}{1+\delta} \mathbb{E}[p_i y_i x_i]^\top \mathbb{E}[R_{-i}] \Sigma w_*$$

$$\simeq \frac{c^\top \bar{R} \Sigma w_*}{1+\delta}.$$

**Variance.** On the other hand, observe that

$$\|\Sigma^{1/2}w_*\|^4 \alpha^2 = (\hat{w}^\top \Sigma w_*)^2 = \hat{w}^\top \Sigma w_* w_*^\top \Sigma \hat{w}.$$

So, applying Eqn (49) with $\Sigma$ replaced with the rank one matrix $\Sigma w_* w_*^\top \Sigma$ and $\Sigma'$ replaced with $R\Sigma w_* w_*^\top \Sigma R$, we get

$$\|\Sigma^{1/2}w_*\|^4 \mathbb{E}\alpha^2 = \mathbb{E}[\hat{w}^\top \Sigma w_* w_*^\top \Sigma \hat{w}] \simeq \frac{1}{(1+\delta)^2}\mathbb{E}[c^\top R\Sigma w_* w_*^\top \Sigma Rc] \simeq \frac{1}{(1+\delta)^2}(c^\top \bar{R}\Sigma w_*)^2,$$

where we have ignored all trace terms which are now of order $1/n$ (negligible). The RHS of the above display coincides with the square of the estimate for $\|\Sigma^{1/2}w_*\|^2 \mathbb{E}[\alpha]$ provided earlier. We deduce that the variance of $\alpha$ vanishes, and so

$$\alpha \simeq \mathbb{E}\alpha \simeq \frac{c^\top \bar{R}\Sigma w_*}{(1+\delta)\|\Sigma^{1/2}w_*\|^2} =: \frac{m}{\|\Sigma^{1/2}w_*\|}.$$

Combining with 48 and Eqn (49) completes the proof of the first part of Proposition 2, namely the convergence Eqn (42).

### E.3  ASYMPTOTICS OF CLASSIFICATION TEST ERROR

In the asymptotic limit Eqn (4), one may use the representation Eqn (42) to write

$$\begin{aligned}
\lim E_{test}(\hat{w}) &= \lim \mathbb{P}(yx^\top \hat{w} \leq 0) \\
&= \mathbb{P}(m|G_1| + \sqrt{\nu - m^2}G_2 \leq 0) \\
&= \mathbb{P}\left(\frac{G_2}{|G_1|} \leq -\frac{m}{\sqrt{\nu - m^2}}\right) \\
&= \mathbb{P}\left(\frac{G_2}{G_1} \leq -\frac{|m|}{\sqrt{\nu - m^2}}\right) \\
&= \frac{1}{2} + \frac{1}{\pi}\arctan(-|m|/\sqrt{\nu - m^2}) \\
&= \frac{1}{\pi}\arccos(|m|/\sqrt{\nu}) = \frac{1}{\pi}\arccos(|m_0|/\sqrt{\nu_0}),
\end{aligned}$$

as claimed. Note that, we have used the fact that $G_2/G_1$ is standard Cauchy random variable, for independent $G_1, G_2 \sim \mathcal{N}(0,1)$. This completes the proof Proposition 2. $\square$

## F  PROOF OF PROPOSITION 1

Using Theorem 4 of (Liao & Mahoney, 2021) (and the proof thereof) combined with some basic algebraic manipulations, we can write

$$R \simeq \bar{R}, \tag{50}$$

$$\text{where } \bar{R}^{-1} = C^{1/2}\mathbb{E}\left[\frac{p_i}{1 + p_i\delta(z)}(\Pi^\perp + (\Pi x_i)(\Pi x_i)^\top)\right]C^{1/2} - zI_d, \tag{51}$$

for a random training example $(x_i, y_i) \sim P_g$ from the generator, and corresponding prune/no prune bit $p_i$. The matrix $C$ is the covariance matrix of $x_i$. Since $p_i$ is Bernoulli with mean $p := \mathbb{P}(p_i = 1)$, it is clear that

$$\mathbb{E}\left[\frac{p_i}{1 + p_i\delta(z)}\right] = \frac{p}{1 + \delta(z)} := t(z).$$

This further gives

$$\bar{R}^{-1} = t(z)C^{1/2}\Pi^\perp C^{1/2} - zI_d + C^{1/2}\Pi K\Pi C^{1/2},$$

$$\text{with } K := \mathbb{E}\left[\frac{p_i}{1 + p_i\delta(z)}uu^\top\right], \tag{52}$$

where $u := \Sigma^{-1/2}x_i \sim \mathcal{N}(0, I_d)$ and $v := C^{1/2}w_o$.

Now, to determine the matrix $K$, we first rewrite $u = (u_{/\!/}, u_\perp)$ and $v = (v_1, v_\perp)$, where

$$u_{/\!/} := \frac{u^\top w_o}{\|w_o\|} \in \mathbb{R}, \quad v_1 := \frac{v^\top w_o}{\|w_o\|} \in \mathbb{R}, \tag{53}$$

$$u_\perp := \Pi^\perp u \in \mathbb{R}^{d-1}, \quad v_\perp := \Pi^\perp v \in \mathbb{R}^{d-1}. \tag{54}$$

The advantage of this representation is that:

- $u_\perp$ and $v_\perp$ are orthogonal to $w_o$.
- $u_{/\!/}$ and $u_\perp$ are statistically independent.
- $u_{/\!/}$ has distribution $\mathcal{N}(0, 1)$.
- $u_\perp$ has distribution $\mathcal{N}(0, I_{d-1})$.

Combining with the fact that due to the evenness of the pruning function $q$ (in Eqn (5), Eqn (6), etc.), the distribution of $(x_i, y_i, p_i)$ doesn't change if $x_i$ is replaced by $-x_i$ (so that $\mathbb{E}[p_i u_i u_j] = 0$ for all $i \neq j$), we get:

$$K = s(z)\Pi + s_\perp(z)\Pi^\perp,$$

$$\text{where } s(z) := \mathbb{E}[h_i G_1^2], \quad s_\perp(z) := \mathbb{E}[h_i G_\perp^2]$$

$$h_i := \frac{p_i}{1 + p_i \delta(z)}, \quad (G_1, G_\perp) \sim \mathcal{N}(0, I_2).$$

Combining with Eqn (52), we get

$$\bar{R}^{-1} = C^{1/2}(a(z)I_d + b(z)\Pi)C^{1/2}, \tag{55}$$

$$\text{where } a(z) = t(z) - z, \quad t(z) = \frac{p}{1 + \delta(z)}, \quad b(z) = s(z) - t(z). \tag{56}$$

Now, using the *Matrix-Inversion Lemma*, one can obtain $\bar{R}$ from $\bar{R}^{-1}$ as follows:

$$C^{1/2}\bar{R}C^{1/2} = (a(z)I_d + b(z)\Pi)^{-1} = \frac{1}{a(z)}\left(I_d - \frac{b(z)/a(z)}{b(z)/a(z) + 1}\Pi\right) = \frac{1}{a(z)}\Pi^\perp + \frac{1}{b(z) + a(z)}\Pi.$$

It suffices to notice that $1/(b(z) + a(z)) = 1/(s(z) - z) = \tilde{m}(z)$ and $1/a(z) = \check{m}(z)$ by definition, and the result follows. $\qquad\square$

## G    Proof of Theorem 1, Theorem 3, and Corollaries

Theorem 1 and Theorem 3 are direct consequences of Proposition 2, where we use the deterministic equivalents provided in Corollary 3, to considerably simplify the resulting formulae. Corollary 1 is a consequence of Theorem 1 and limiting behavior of the spectral functions given in Eqn 37.

### G.1    Proof of Theorem 1 and Theorem 3

Set $z = -\lambda$. Also recall that $c = \beta_1 u + \beta_2 v$, where $u$, $v$, $\beta_1$, and $\beta_2$ are as in Lemma 3. Note that we have the identification $\beta = \beta_2$ and $\tilde{\beta} = \beta_1$. We know from Proposition 1 that $R \simeq \bar{R} = m(z)\Pi^\perp + \tilde{m}(z)\Pi$, where $\Pi = uu^\top$. One computes

$$m_0 = (w_*/\|w_*\|)^\top \bar{R}c = \frac{1}{\|w_*\|}w_*^\top \left(m(z)\Pi^\perp + \tilde{m}(z)\Pi\right)(\beta_1 u + \beta_2 v),$$

$$= \frac{1}{\|w_*\|}w_*^\top \left(\beta_1 \tilde{m}(z)u + \beta_2 m(z)v\right),$$

Moreover, on computes $w_*^\top u/\|w_*\| = \rho_*$ by definition, and

$$\frac{w_*^\top v}{\|w_*\|} = \frac{(w_g - (w_g^\top w_o)w_o)^\top w_*/\|w_*\|}{\|w_g - (w_g^\top w_o)w_o\|} = \frac{w_g^\top w_*/\|w_*\| - \rho_g\|w_g\|(w_o^\top w_*/\|w_*\|)}{\|w_g\|\sqrt{1 - \rho_g^2}}$$

$$= \frac{\rho - \rho_g\rho_*}{\sqrt{1 - \rho_g^2}} = \frac{\cos\theta - \cos\theta_g \cos\theta_*}{\sin\theta_g} = \sin\theta_* \cos\xi = \sqrt{1 - \rho_*^2}\cos\xi =: \omega/\beta_2,$$

where we have used the identity $\cos\theta = \cos\theta_g\cos\theta_* + \sin\theta_g\sin\theta_*\cos\xi$, known as the *Spherical Law of Cosines*. Putting things together gives $m_0 \simeq \omega m(z) + \tilde{\omega}\tilde{m}(z)$ as claimed.

Likewise, one computes

$$\frac{1}{n}\operatorname{tr}\Sigma C' = \frac{1}{n}\operatorname{tr}R^2 \simeq \frac{1}{n}\operatorname{tr}\left(m'(z)\Pi^\perp + \tilde{m}'(z)\Pi\right) \simeq \phi m'(z),$$
$$c^\top \bar{R}c = c^\top\left(m(z)\Pi^\perp + \tilde{m}(z)\Pi\right)c = (\beta_1 u + \beta_2 v)^\top(\tilde{m}(z)\Pi + m(z)\Pi^\perp)(\beta_1 u + \beta_2 v)$$
$$= \beta_2^2 m(z) + \beta_1^2\tilde{m}(z) = \beta^2 m(z) + \tilde{\beta}^2\tilde{m}(z) =: r(z),$$
$$c^\top\Sigma'c = c^\top\mathbb{E}\left[R^2\right]c \simeq c^\top\left(m'(z)\Pi^\perp + \tilde{m}'(z)\Pi\right)c = \beta^2 m'(z) + \tilde{\beta}^2\tilde{m}'(z) = r'(z).$$

We deduce that $\nu = \nu_0/(1+\delta)^2$, where

$$\nu_0 = \frac{p}{n}\operatorname{tr}\Sigma C' + c^\top\Sigma'c - \frac{2c^\top\bar{R}c}{1+\delta}\frac{1}{n}\operatorname{tr}C\Sigma'$$
$$\simeq \frac{p}{n}\operatorname{tr}R^2 + r'(z) - \frac{2r(z)}{1+\delta(z)}\frac{1}{n}\operatorname{tr}R^2 = p\phi m'(z) + r'(z) - \frac{2r(z)\phi m'(z)}{1+\phi m(z)}.$$

the result then follows from Proposition 2. $\qquad\square$

## G.2 PROOF OF COROLLARY 1

As usual, set $z := -\lambda < 0$.

(A) For $\phi < p$, it is easy to see from formula Eqn (33) and Lemma 2 that in the limit $z \to 0$, one has

$$m(z) \to \frac{1}{p-\phi},$$
$$\bar{m}(z) \to 0,$$
$$\tilde{m}(z) \to \frac{p/\gamma}{p-\phi},$$
$$m'(z) \to \frac{p}{(p-\phi)^3},$$
$$\bar{m}'(z) \to \frac{1}{p-\phi},$$
$$\tilde{m}'(z) \to \frac{p/\gamma^2}{(p-\phi)^3}\left(p(p-\phi)+\phi\gamma\right) = \frac{p}{(p-\phi)^3}\left((p-\phi)p/\gamma^2 + \phi/\gamma\right),$$
$$\frac{m'(z)}{1+\phi m(z)} \to \frac{1}{(p-\phi)^2}.$$

Furthermore, with $m_0$ and $\nu_0$ as defined in Theorem 1, one computes

$$r(z) = \beta^2 m(z) + \tilde{\beta}^2\tilde{m}(z) \to \beta^2\frac{1}{p-\phi} + \tilde{\beta}^2\frac{p/\gamma}{p-\phi} = \frac{r_0}{p-\phi},$$
$$r'(z) = \beta^2 m'(z) + \tilde{\beta}^2\tilde{m}'(z) \to \beta^2\cdot\frac{p}{(p-\phi)^3} + \tilde{\beta}^2\cdot\frac{p/\gamma^2}{(p-\phi)^3}(p(p-\phi)+\phi\gamma) = \frac{r_0'}{(p-\phi)^3},$$

where $r_0$ and $r_0'$ are as defined in the claim. We deduce that $m_0/\sqrt{\nu_0 - m_0^2} = a/\sqrt{b-a^2}$ and the result follows from Theorem 1.

(B) Now consider the case $\phi > p$. Observe that $m_0 = \sqrt{\nu_0 - m_0^2} = -zm_0/\sqrt{z^2 - z^2 m_0^2}$. On the other hand, from Eqn (33) we know that

$$-zm(z) = \frac{\sqrt{(p-\phi-z)^2 - 4\phi z} - (p-\phi-z)}{2\phi} \tag{57}$$

Combining with Lemma 2, we deduce the following limits

$$-zm(z), z^2 m'(z) \to c_0 := 1 - p/\phi > 0,$$

$$\bar{m}'(z) \to \frac{p/\phi}{\phi - p},$$

$$-z\tilde{m}(z), z^2 \tilde{m}'(z) \to \frac{c_0}{\gamma/\phi + c_0},$$

$$\frac{-zm'(z)}{1 + \phi m(z)} \to \frac{1}{\phi}.$$

Furthermore, one computes

$$-zr(z) = \beta_2^2 \cdot (-zm(z)) + \beta_1^2 \cdot (-z\tilde{m}(z)) = \beta_2^2 c_0 + \beta_1^2 \frac{c_0}{\gamma/\phi + c_0} =: c_0 r_0,$$

$$z^2 r'(z) = \beta_2^2 z^2 m'(z) + \beta_1^2 z^2 \tilde{m}(z) = \beta_2^2 c_0 + \beta_1^2 \frac{c_0}{\gamma/\phi + c_0} = c_0 r_0,$$

$$-zm_0 = \sqrt{2/\pi} \cdot (-zm(z)\omega - z\tilde{m}(z)\tilde{\omega}) \to \sqrt{2/\pi} c_0 \cdot (\omega + \tilde{\omega}/(\gamma/\phi + c_0)) := a,$$

$$z^2 \nu_0 = p\phi z^2 m'(z) + z^2 r'(z) - 2\phi \frac{-zm'(z)}{1 + \phi m(z)} \cdot (-zr(z))$$

$$\to p\phi c_0 + r_0 c_0 - 2r_0 c_0 = c_0 \cdot (p\phi - r_0) =: b.$$

We deduce that

$$m_0/\sqrt{\nu_0} = -za/\sqrt{z^2 b} = a/\sqrt{b},$$

and the result follows from Theorem 1. □

### G.3 PROOF OF THEOREM 2

Taking the limit $\phi \to 0$ in Corollary 1, we have

$$r_0' \to p \cdot (\beta^2 + \tilde{\beta}^2 p^2/\gamma^2), \quad b \to \frac{\beta^2 + \tilde{\beta}^2 p^2/\gamma^2}{p^2}, \quad a \to \frac{\omega + \tilde{\omega}p/\gamma}{p},$$

$$a/\sqrt{b} \to \frac{\omega/p + \tilde{\omega}/\gamma}{\sqrt{\beta^2/p^2 + \tilde{\beta}^2/\gamma^2}} = \frac{(\beta/p)\sqrt{1 - \rho_*^2} \cos\zeta + (\tilde{\beta}/\gamma)\rho_*}{\sqrt{\beta^2/p^2 + \tilde{\beta}^2/\gamma^2}} = \frac{j\sqrt{1 - \rho_*^2} \cos\zeta + 1}{\sqrt{j^2 + 1}},$$

with $j = j(q) := \frac{\gamma(q)\beta(q)}{p\tilde{\beta}(q)} > 0.$

where we recall that $\omega = \beta\sqrt{1 - \rho_*^2} \cos\zeta$ and $\tilde{\omega} = \tilde{\beta}\rho_*$.

**Part (A).** Taking $\rho_* = 1$, meaning that pruning is done along the ground-truth, gives

$$a/\sqrt{b} = 1/\sqrt{j^2 + 1}.$$

From Corollary 1, we see that the limiting value of $E_{clf}(\hat{w})$, i.e the functional $F$ defined in Eqn (12), is an increasing function of the ratio $j(q)$. The proof is completed by invoking Lemma 5 which establishes that i$q_{\text{KH}(p)}$ (resp. $q_{\text{KE}(p)}$) is the unique minimizer (resp. maximizer) of the ratio $j(q)$ over $q \in \mathcal{Q}_p$.

**Part (B).** On the other hand, taking $\rho = 1$ gives $\rho_g = \rho_*$, $\zeta = 0$, $\omega = \beta\sqrt{1 - \rho_g^2}$. We get $a > 0$, and

$$a/\sqrt{b} \to \frac{j\sqrt{1 - \rho_*^2} + \rho_*}{\sqrt{j^2 + 1}}.$$

It is easy to show that the RHS is strictly decreasing function of $j$. As with part (A), the proof is completely by invoking Lemma 5 to extremize the ratio $j = j(q)$. □

**Lemma 5.** *Suppose $\rho_g > 0$. For any fixed pruning strategy $p \in (0, 1]$, ignoring null-sets, the unique maximizer (resp. minimizer) of the ratio $j(q)$ over $\mathcal{Q}_p := \{q \in \mathcal{Q} \mid p(q) = p\}$ is the "keep hard examples" pruning strategy $q_{\text{KH}(p)}$ (resp. the "keep easy examples" pruning strategy $q_{\text{KE}(p)}$).*

*Proof.* Clearly, there is a bijective correspondence between $\mathcal{Q}_p$ and the collection $\mathcal{S}_p$ of Borell subsets $S \subseteq \mathbb{R}$ of Gaussian measure equal to $p$, and verifying the symmetry condition $-S = S$. This correspondence is simply $S \mapsto 1_S$, the indicator function of $S$. Furthermore, for any $S \in \mathcal{S}_p$, one can write

$$\gamma(1_S) = 2F_0(S_+), \quad \tilde{\beta}(1_S) = 2F_1(S_+), \quad \beta(1_S) = 2F_2(S_+), \text{ with}$$

$$S_+ := S \cap (0, \infty), \quad F_k(T) := \int_T f_k(t)\varphi(t)\mathrm{d}t,$$

$$f_0(t) := t^2, \quad f_1(t) := (2\Phi(\tau t) - 1)t, \quad f_2(t) := \varphi(\tau t), \quad \tau := \rho_g/\sqrt{1 - \rho_g^2}.$$

Define $a_p, b_p > 0$ such that the sets $I_p := \{t \in \mathbb{R} \mid |t| \geq a_p\}$ and $J_p := \{t \in \mathbb{R} \mid |t| \leq b_p\}$ both have Gaussian measure $p$. We shall show that over the collection $\mathcal{T}_p$ of Borell subsets of $(0, \infty)$ with Gaussian measure equal to $m = p/2$, the functional $T \mapsto F_0(T)F_2(T)$ is minimized (resp. maximized) by $J_p^+ := [a_p, \infty)$ (resp. $I_p^+ := [0, b_p]$), while modulo null sets, and $F_1$ is uniquely maximized (resp. minimized) by $J_p^+$ (resp. $I_p^+$).

**Step 1: Reduction to Integration w.r.t Lebesgue Measure.** For any $t > 0$ and $u \in [0, 1/2]$, define

$$M(t) := \mu([0, t]), \quad N(u) := M^{-1}(u).$$

Under the change of variable $t = N(u)$, one has

$$F_k(T) = \bar{F}_k(M(T)), \text{ where } \bar{F}(U) := \int_U g_k(u)\mathrm{d}u, \quad g_k := f_k \circ N, \text{ and } M(T) := \{M(t) \mid t \in T\}.$$

Thus, the minimizers (resp. maximizers) of $F$ over $T \in \mathcal{T}_p$ are of the form $N(U)$ where $U$ minimizes (resp. maximizes) $\bar{F}(U) := \bar{F}_0(U)\bar{F}_1(U)/\bar{F}_2(U)$ over Borell sets $U \subseteq (0, 1/2)$ verifying $|U| = m$. Let us show that modulo null sets, $\bar{F}$ is minimized by $(0, m]$ and maximized by $(1/2 - m, 1/2)$ where $m := p/2 \in (0, 1/2)$.

For any $r \geq 0$, consider the equivalent linear-fractional program

$$\min_{r \geq 0, U \subseteq (0, 1/2)} \frac{r\bar{F}_1(U)}{\bar{F}_2(U)} \text{ subject to } |U| = m, \bar{F}_0(U) \leq r. \tag{58}$$

**Step 2: Dinkelback re-Parametrization.** For fixed $r \geq 0$, consider the change of variable $\lambda = \bar{F}_1(U)/\bar{F}_2(U)$, and define

$$v(\lambda) := \max_{U \subseteq (0, 1/2)} \bar{F}_1(U) - \lambda \bar{F}_2(U) \text{ subject to } |U| = m, \bar{F}_0(U) \leq r. \tag{59}$$

The "Dinkelbach trick" tells us that $\lambda^* = \max_U \bar{F}_1(U)/\bar{F}_2(U)$ iff $v(\lambda^*) = 0$.

Now, the Lagrangian for the auxiliary problem is given by

$$\mathcal{L}(U, \lambda, \eta, \zeta) = \bar{F}_1(U) - \lambda\bar{F}_2(U) + \eta \cdot (r - \bar{F}_0(U)) + \zeta \cdot (m - |U|)$$

$$= \int_U H(u, \lambda, \eta, \zeta)\mathrm{d}u + \eta r + \zeta m, \text{ with } H(u, \lambda, \eta, \zeta) := g_1(u) - \lambda g_2(u) - \eta g_0(u) - \zeta.$$

The first-order optimality conditions of $U$ can then be expressed as

$$H(u, \lambda, \eta, \zeta) \begin{cases} \geq 0, & \text{if } u \in U, \\ \leq 0, & \text{otherwise.} \end{cases} \tag{60}$$

**Step 3: Shape Analysis.** Now, under the assumption that $\rho_g > 0$, the functions $f_0$ and $f_1$ (therefore $g_0$ and $g_2$) are increasing and $g_1$ (therefore $g_1$) is decreasing. Thus, for any $\lambda, \eta \geq 0$, the function $u \mapsto H(u, \lambda, \eta, \zeta)$ is a non-increasing function, for any feasible $\lambda, \eta, \zeta$. A non-increasing function crosses zero at most once. We deduce that the optimal $U$ must be of the form $[b, 1/2)$, modulo a null set. The condition $|U| = m$ forces $b = 1/2 - m$. We conclude that $[1/2 - m, 1/2)$ is the unique minimizer of $\bar{F}$.

Similarly, one shows that $[0, m]$ is the unique maximizer of $\bar{F}$. $\qquad\square$

# H    PROOF OF THEOREM 4 (REGRESSION ANALYSIS)

## H.1    A MODIFIED BIAS-VARIANCE DECOMPOSITION

We start with the following general bias-variance decomposition for the regression test error.

**Proposition 3.** *The regression test error of the estimator $\hat{w}$ defined in Eqn Eqn (3) is given exactly by*

$$E_{reg}(\hat{w}) = \lambda^2 \mathbb{E}\left[w_g^\top R \Sigma R w_g\right] + \sigma^2 \mathbb{E}\frac{1}{n} \operatorname{tr} S R^2 \Sigma + c^2 - 2\lambda \mathbb{E}\left[w_g^\top R \Sigma \epsilon\right], \tag{61}$$

*where $\epsilon := w_g - w_*$, $c^2 := \epsilon^\top \Sigma \epsilon$, and $S$ and $R$ are the random matrices defined in Eqn Eqn (3).*

The first two terms in the above sum correspond to bias and variance if we had $w_g = w_*$, i.e if we had no label-shift; the last two terms in red are a correction to take into account label shift.

## H.2    PROOF OF THEOREM 4

Now, from Proposition 1 with $\Sigma = I_d$, we have the following deterministic equivalents:

$$R \simeq m(z)\Pi^\perp + \tilde{m}(z)\Pi,$$

$$SR - I_d = zR \simeq zm(z)\Pi^\perp + z\tilde{m}(z)\Pi,$$

$$R^2 = \frac{\partial}{\partial z}R \simeq m'(z)\Pi^\perp + \tilde{m}'(z)\Pi,$$

$$SR^2 = \frac{\partial}{\partial z}SR \simeq (m(z) + zm'(z))\Pi^\perp + (\tilde{m}(z) + z\tilde{m}'(z))\Pi$$

$$= (m(z) + zm'(z))I_d + (\tilde{m}(z) - m(z) + z\tilde{m}'(z) - zm'(z))\Pi.$$

Furthermore, notice that because $\Pi$ is a fixed-rank (in fact rank-1) matrix, so is $S\Pi\Sigma$, and so $\mathbb{E}(1/n)\operatorname{tr}S\Pi\Sigma \to 0$ in the limit $n \to \infty$. Thus, in view of using Proposition 3, one computes

$$\mathbb{E}\left[w_g^\top R \Sigma R w_g\right] = w_g^\top \mathbb{E}\left[R^2\right]w_g = m'(z)\|w_g^\perp\|^2 + \tilde{m}'(z)\|w_g^\varnothing\|^2,$$

$$\mathbb{E}\frac{1}{n}\operatorname{tr}SR^2\Sigma \simeq \phi \cdot \mathbb{E}\frac{1}{d}\operatorname{tr}SR^2\Sigma \simeq \phi \cdot (m(z) + zm'(z)) = \phi\bar{m}'(z),$$

$$\mathbb{E}\left[w_g^\top R \Sigma \epsilon\right] = \mathbb{E}\left[w_g^\top R \epsilon\right] \simeq \epsilon^\top(m(z)w_g^\perp + \tilde{m}(z)w_g^\varnothing).$$

Putting things together then gives

$$E_{reg}(\hat{w}) \simeq \lambda^2 \cdot \left(m'(-\lambda)\|w_g^\perp\|^2 + \tilde{m}'(-\lambda)\|w_g^\varnothing\|^2\right) + \sigma^2\phi\bar{m}'(-\lambda)$$

$$+ \|\epsilon\|^2 - 2\lambda\epsilon^\top(m(-\lambda)w_g^\perp + \tilde{m}(-\lambda)w_g^\varnothing)$$

$$= \lambda^2 \cdot \left(m'(-\lambda)\|w_g^\perp\|^2 + \tilde{m}'(-\lambda)\|w_g^\varnothing\|^2\right) + \sigma^2\phi\bar{m}'(-\lambda)$$

$$+ c^2 - 2\lambda \cdot (m(-\lambda)a + \tilde{m}(-\lambda)b) \text{ with } a := \epsilon^\top w_g^\perp, \, b := \epsilon^\top w_g^\varnothing \text{ and } c^2 := \|\epsilon\|^2,$$

which proves Theorem 4. $\qquad\qquad\square$

## H.3    PROOF OF COROLLARY 2

The first equation follows by taking the limit $\phi \to 0^+$ in part (A) of Corollary 1. For the second equation, note that in the limit Eqn (4) Corollary 1 gives $E_{reg} \simeq L = c^2 + L_0$, with

$$L_0 = L_0(\phi, p) := \begin{cases} 0, & \text{if } \phi < p, \\ c_0 D + \frac{c_0}{c_1}E, & \text{if } \phi > p, \end{cases}$$

where $D := \|w_g^\perp\|^2 - 2a$, $E := \|w_g^\varnothing\|^2 - 2b$, and we recall that

$$c_0 := 1 - p/\phi, \quad c_1 := \gamma/\phi + c_0 = 1 - (p - \gamma)/\phi, \quad \gamma = p + 2\alpha\varphi(\alpha), \quad \alpha = \Phi^{-1}(1 - p/2).$$

Now, on the second branch, one computes

$$\gamma' := \frac{\partial \gamma}{\partial p} = \alpha^2, \quad \frac{\partial L_0}{\partial p} = -\frac{D}{\phi} - E\frac{\gamma + (\phi - p)\gamma'}{(\phi - (p - \gamma))^2} = -\frac{D}{\phi} - E\frac{\gamma + (\phi - p)\alpha^2}{(\phi - (p - \gamma))^2},$$

One can further show the Hessian of $L_0$ is nonnegative everywhere provided $E > 0$, and so every stationary point is a global minimum, provided it lies in the interval $(0, \phi)$. Expanding to first order in $p$, observe that if $t := -D/E > 0$, then we have a unique stationary point $p_0 = p_0(\phi)$. By the way, observe that $D + c^2 = \|w_* - w_g^/\|^2$ and $E + c^2 = \|w_* - w_g^\perp\|^2$, where $c^2 := \|w_* - w_g\|^2$ as usual, and so the condition $D < 0 < E$ is equivalent to the condition $\|w_* - w_g^/\|^2 < c^2 < \|w_* - w_g^\perp\|^2$ in the statement of the result being proved. Further, one can show that for small $\phi$,

$$p_0/\phi \simeq \sqrt{t}/\sqrt{2\log 1/\phi}, \quad (\gamma(p_0) - p_0)/\phi \simeq \sqrt{t}\sqrt{2\log 1/\phi}. \tag{62}$$

See Lemma 6. It is clear that $p_0 \ll \phi$ because $\log 1/\phi \gg 1$ for small $\phi$, and so $p_0$ is on the second branch of the definition of $L_0(\phi, p)$, and must therefore be a global min of $L_0$ the interval $(0, \phi)$.

Moreover, one has (still in the limit $\phi \to 0^+$)

$$\log 1/\phi \to \infty, \quad c(p_0) = 1 - p_0/\phi \to 1, \quad c_1(p_0) = 1 + (\gamma(p_0) - p_0)/\phi \to \infty, \quad c_0(p_0)/c_1(p_0) \to 0,$$

and so $\lim_{\phi \to 0^+} L(\phi, p_0(\phi)) = c^2 + \lim_{\phi \to 0^+} L_0(\phi, p_0(\phi)) = D + c^2 = \|w_* - w_g^/\|^2 < c^2$. $\square$

**Lemma 6.** *Let $t$ and $p_0$ be as in the proof of Corollary 2. For $\phi \to 0^+$, it holds that*

$$p_0 \simeq \sqrt{t}/\sqrt{2\log 1/\phi}, \quad (\gamma(p_0) - p_0)/\phi \simeq \sqrt{t}\sqrt{2\log 1/\phi}. \tag{63}$$

*Proof.* The idea is to argue that $p$ must be small, and so we must have $\alpha$ large and $\gamma \gg 0$. One then considers the simplified equation $D \cdot (\phi + \gamma(p))^2 + E\phi^2\alpha(p)^2 = 0$, which can be solved as a function $p_0(\phi)$ of $\phi$ using Lambert-W function. Finally, since $\phi$ is small $p_\phi$, we can further drop the Lambert-W function and ultimately get $p_0 \simeq \sqrt{t}/\sqrt{2\log 1/\phi}$. $\square$

### H.4 PROOF OF PROPOSITION 3

As usual, set $z := -\lambda$ so that $R = (S - zI_d)^{-1}$. Observe that the estimator given in Eqn Eqn (3) can be written as $\hat{w} = RSw_g + RX^\top D\Delta/n$, where $\Delta := Y - Xw_g \in \mathbb{R}^n$ is the vector of epistemic label noise, which is independent of the design matrix $X$, and has distribution $\mathcal{N}(0, \sigma^2 I_n)$. We may then decompose the regression test error of $\hat{w}$ as follows:

$$
\begin{aligned}
E_{reg}(\hat{w}) &= \mathbb{E}\left[(x^\top\hat{w} - y)^2\right] - \sigma^2 = \mathbb{E}\left[(x^\top\hat{w} - x^\top w_*)^2\right] = \mathbb{E}\left[\|\hat{w} - w_*\|_\Sigma^2\right] \\
&= \mathbb{E}\left[\|RSw_g + RX^\top D\Delta/n - w_*\|_\Sigma^2\right], \\
&= \mathbb{E}\left[\|RSw_g - w_*\|_\Sigma^2\right] + \mathbb{E}\left[\|RX^\top D\Delta/n\|_\Sigma^2\right], \\
&= \mathbb{E}\left[\|RSw_g - w_g + w_g - w_*\|_\Sigma^2\right] + \sigma^2\mathbb{E}\frac{1}{n^2}\operatorname{tr} DXR\Sigma RX^\top D \\
&= \mathbb{E}\left[\|RSw_g - w_g\|_\Sigma^2\right] + \sigma^2\mathbb{E}\frac{1}{n}\operatorname{tr} SR^2\Sigma + \operatorname{tr}\Sigma\Delta + 2\mathbb{E}\left[w_g^\top(SR - I_d)\Sigma\epsilon\right] \\
&= z^2\mathbb{E}\left[w_g^\top R\Sigma Rw_g\right] + \sigma^2\mathbb{E}\frac{1}{n}\operatorname{tr} SR^2\Sigma + \epsilon^\top\Sigma\epsilon + 2z\mathbb{E}\left[w_g^\top R\Sigma\epsilon\right],
\end{aligned}
$$

where we have used the elementary identity $SR - I_d = zR$. $\square$

## I  PROOF OF THEOREM 5 (OPTIMAL PRUNING IN REGRESSION SETTING)

Note that the pruning strategy $q$ only enters the picture via the parameter $p(q) := \mathbb{E}[q(G)]$ and $\gamma(q) := \mathbb{E}[q(G)G^2]$.

**Definition 2.** *Let $\mathcal{Q}$ be the set of all admissible pruning strategies satisfying Assumption 1, and for any subset of $\mathcal{H}$ of $\mathcal{Q}$, define $\operatorname{Spec}(\mathcal{H}) \subseteq [0,1]^2$ as follows:*

$$\operatorname{Spec}(\mathcal{H}) := \{(p(q), \gamma(q)) \mid q \in \mathcal{H}\}. \tag{64}$$

*Thus, $\operatorname{Spec}(\mathcal{H})$ collects all possible values of $p$ and $\gamma$ attainable by some pruning strategy $q \in \mathcal{H}$.*

Let $\mathcal{Q}_* := \{q_{p,u} \mid (p,u) \in [0,1]^2\} \subseteq \mathcal{Q}$, where $q_{p,u}$ is as defined in Eqn (21). The next result gives us a tractable description of $\mathrm{Spec}(\mathcal{Q})$. In particular, it proves Theorem 5.

**Proposition 4.** *We have the following analytic descriptions for* $\mathrm{Spec}(\mathcal{Q})$:

$$\mathrm{Spec}(\mathcal{Q}) = \mathrm{Spec}(\mathcal{Q}_*), \tag{65}$$

$$\mathrm{Spec}(\mathcal{Q}) = \{(p,\gamma) \mid 0 \le p \le 1, \gamma_{min}(p) \le \gamma \le \gamma_{max}(p)\}, \tag{66}$$

*where* $\gamma_{min}(p) := p - 2\alpha_{min}(p)\varphi(\alpha_{min}(p)), \quad$ *with* $\alpha_{min}(p) := \Phi^{-1}((1+p)/2),$ (67)

$$\gamma_{max}(p) := p + 2\alpha_{max}(p)\varphi(\alpha_{max}(p)), \quad \text{with } \alpha_{max}(p) := \Phi^{-1}(1-p/2). \tag{68}$$

*Geometrically,* $\mathrm{Spec}(\mathcal{Q})$ *is thus the lens-like region between graphs of the functions* $\gamma_{min}$ *and* $\gamma_{max}$.

*Proof.* Recall the functions $\alpha_{min}(p) := \Phi^{-1}((1+p)/2)$, $\alpha_{max}(p) := \Phi^{-1}(1-p/2)$, $\gamma_{min}(p) := p - 2\alpha_{min}(p)\varphi(\alpha_{min}(p))$ and $\gamma_{max}(p) := p + 2\alpha_{max}(p)\varphi(\alpha_{max}(p))$ introduced in the lemma.

First note that any $q \in \mathcal{Q}$ is the indicator function of a disjoint union of intervals $A = \cup_{I \in \mathcal{I}} I$ such that $I \in \mathcal{I}$ iff $-I \in \mathcal{I}$, where $-I := \{-t \mid t \in I\}$. Now, for any $p \in [0,1]$, the minimum (resp. maximum) feasible value for $\gamma(q)$ over the surface $\{q \in \mathcal{Q} \mid p(q) = p\}$ is $\gamma_{min}(p)$ (resp. $\gamma_{max}(p)$) and it is attained by taking the "keep easy" pruning strategy $q(t) := 1_{|t| \le \alpha_{min}(p)}$ (resp. "keep hard" pruning strategy $q(t) := 1_{|t| \ge \alpha_{max}(p)}$). See Lemma 7. Therefore, we must have

$$\mathrm{Spec}(\mathcal{Q}) := \{(p(q), \gamma(q)) \mid q \in \mathcal{Q}\} \subseteq \{(p,\gamma) \mid p \in [0,1], \gamma \in \Gamma(p)\},$$

where we recall that $\Gamma(p) := [\gamma_{min}(p), \gamma_{max}(p)]$.

We now show the other direction of the set inclusion above. Given $\gamma \in \Gamma(p)$, we must construct $q \in \mathcal{Q}$ such that $p(q) = p$ and $\gamma(q) = \gamma$. Indeed, for any $u \in [0,1]$, define $q_u \in \mathcal{Q}$ as the indicator function of the union of the intervals $I_u := \{t \in \mathbb{R} \mid |t| \le a(u)\}$ and $J_u := \{t \in \mathbb{R} \mid |t| > b(u)\}$, where $a(u) := \alpha_{min}((1-u)p)$ and $b(u) := \alpha_{min}(pu)$. It is easy to verify that $b(u) \ge a(u)$. Indeed, because $\Phi^{-1}$ is non-decreasing, we know from the definition of $\alpha_{max}$ and $\alpha_{min}$ functions that

$$\alpha_{max}(pu) \ge \alpha_{min}((1-u)p) \iff 1-pu/2 \ge (1+(1-u)p)/2 \iff (1+p)/2 \le 1 \iff p \le 1.$$

If follows that $I_u$ and $J_u$ are disjoint and so

$$q_u(t) = 1_{I_u \cup J_u} = 1_{I_u} + 1_{J_u},$$

It is easy to verify that $p(q_u) = pu + (1-u)p = p$ and

$$\gamma(q_u) = p - 2a(u)\varphi(a(u)) + 2b(u)\varphi(b(u)).$$

Observe that $u \mapsto \gamma(q_u)$ increases continuously from $\gamma_{min}(p)$ at $u = 0$ to $\gamma_{max}(p)$ for $u = 1$. It follows from the *Intermediate Value Theorem* that there exists $u_0 \in [0,1]$ such that $\gamma(q_{u_0}) = \gamma$. It suffices to take $q = q_{u_0}$.

Finally, $\mathrm{Spec}(\mathcal{Q}) = \mathrm{Spec}(\mathcal{Q}_*)$ follows directly from the construction of $q_u$. $\square$

**Lemma 7.** *For any $p \in [0,1]$, we have the following.*

*(A) The minimum of $\gamma(q)$ over all $q \in \mathcal{Q}$ is given by*

$$\gamma_{min}(p) = p - \alpha_{min}(p)\varphi(\alpha_{min}(p)), \text{ with } \alpha_{min}(p) := \Phi^{-1}((1+p)/2), \tag{69}$$

*and is attained by setting $q(t) \equiv 1_{|t| \le \alpha_{min}(p)}$.*

*(B) The maximum of $\gamma(q)$ over all $q \in \mathcal{Q}$ is given by*

$$\gamma_{max}(p) = p + \alpha_{max}(p)\varphi(\alpha_{max}(p)), \text{ with } \alpha_{max}(p) := \Phi^{-1}(1-p/2). \tag{70}$$

*and is attained by setting $q(t) = 1_{|t| > \alpha_{max}(p)}$.*

## J PROOFS OF LEMMAS

### J.1 PROOF OF LEMMA 2

The formula for $m'(z)$ from differentiating through Eqn (34) w.r.t $z$, and then doing some basic algebraic manipulations. All the other formulae for $\bar{m}'(z)$, $\tilde{m}(z)$, and $r'(z)$ follow from the definition of the quantities and the chain rule. $\square$

## K    PROOF OF LEMMA 7

(A) Every $q \in \mathcal{Q}$ is the indicator function of some measurable $A \subseteq \mathbb{R}$. We wish to maximize $\gamma(q) = \int_A t^2 \varphi(t) \mathrm{d}t$ over $A$, subject to $p(q) = \int_A \varphi(t) \mathrm{d}t = p$. The Lagrangian is

$$\mathcal{L}(A, \lambda) = \int_A t^2 \varphi(t) \mathrm{d}t + \lambda \cdot \left( p - \int_A \varphi(t) \mathrm{d}t \right) = \int_{-\infty}^{\infty} (t^2 - \lambda) 1_A(t) \varphi(t) \mathrm{d}t + p\lambda.$$

Since $\varphi(t) > 0$ for all $t$, it is clear that the integrand is minimized by taking

$$1_A(t) = \begin{cases} 1, & \text{if } t^2 > \lambda \\ 0, & \text{otherwise.} \end{cases}$$

Thus, by the *Rearrangement inequality* (for measures), it is optimal to take $A = (-\infty, \sqrt{\lambda}) \cup (\sqrt{\lambda}, \infty)$ for some $\lambda \geq 0$. The constraint $\int_A \varphi(t) \mathrm{d}t = p$ then gives

$$\sqrt{\lambda} = \Phi^{-1}((1+p)/2) =: \alpha_{min}(p).$$

(B) Analogous arguments. $\qquad\qquad\square$

## L    PROOF OF LEMMA 3

### L.1    NON-LIMO CASE

Let us prove the formula for $\beta_1$ and $\beta_2$ given in the first row of Table 3. Consider $F = \mathrm{sign}(U)q(V)$, where $U = Z^\top \bar{w}_g$ and $V := Z^\top \bar{w}_o$, for $Z \sim \mathcal{N}(0, I_d)$. Note that we can write $C^{-1/2}c = \mathbb{E}[FZ]$. By Stein's lemma, we have $C^{-1/2}c = a\bar{w}_g + b\bar{w}_o$, where

$$a := \mathbb{E}[\frac{\partial F}{\partial U}], \quad b := \mathbb{E}[\frac{\partial F}{\partial V}]. \tag{71}$$

By direct computation, one has

$$\frac{\partial F}{\partial U} = 2\delta(U)q(V), \tag{72}$$

$$\frac{\partial F}{\partial V} = \mathrm{sign}(U)q'(V), \tag{73}$$

in the distribution-theoretic sense. Thus, one computes

$$\mathbb{E}[\delta(U)q(V)] = \varphi(0)\mathbb{E}[q(V) \mid U = 0] = \varphi(0)\mathbb{E}[q(V) \mid U = 0] = \varphi(0)\mathbb{E}[q(G)]$$

$$= \varphi(0) \int_{-\infty}^{\infty} q(\sigma t)\varphi(t) \mathrm{d}t = \frac{\varphi(0)}{\sigma} \int_{-\infty}^{\infty} q(t)\varphi(t/\sigma) \mathrm{d}t$$

$$= \frac{1}{\sigma} \mathbb{E}[q(G)\varphi(\tau G)],$$

where we have used the fact that

$$\varphi(\tau t)\varphi(t) = \frac{1}{\sqrt{2\pi}} \varphi(t\sqrt{\tau^2 + 1}) = \varphi(0)\varphi(t/\sqrt{1-\rho^2}) = \varphi(0)\varphi(t/\sigma).$$

We deduce that $a = (2/\sigma)\mathbb{E}[q(G)\varphi(\tau G)]$.

On the other hand, for any $s \in \mathbb{R}$, one computes

$$\mathbb{E}[\mathrm{sign}(U)\delta(V - s)] = \varphi(s)\mathbb{E}[\mathrm{sign}(U) \mid V = s]$$

$$= \varphi(s)(\mathbb{P}(U \geq 0 \mid V = s) - \mathbb{P}(U < 0 \mid V = s)).$$

But, conditioned on $V = s$ the distribution of $U$ is $\mathcal{N}(\rho_g s, \sigma^2)$, where $\sigma := \sqrt{1 - \rho_g^2}$. We deduce that $\mathbb{P}(U \geq 0 \mid V = s) = \mathbb{P}(\mathcal{N}(0, \sigma^2) \geq -\rho_g s) = \mathbb{P}(\mathcal{N}(0, \sigma^2) \leq \rho_g s) = \Phi(\tau s)$. Likewise, $\mathbb{P}(U < 0 \mid V = s) = \mathbb{P}(\mathcal{N}(0, \sigma^2) < -\rho_g s) = \Phi(-\tau s) = 1 - \Phi(\tau s)$. We deduce that $\mathbb{E}[\mathrm{sign}(U)\delta(V - s)] = \varphi(s)(2\Phi(\tau s) - 1)$, and so

$$\mathbb{E}[\mathrm{sign}(U)q'(V) \mid V = s] = \int q'(s)(2\Phi(\tau s) - 1)\varphi(s) \mathrm{d}x = \mathbb{E}[q'(G)(2\Phi(\tau G) - 1)]$$

$$= 2\mathbb{E}[q'(G)\Phi(\tau G)] - \mathbb{E}[q'(G)] = 2\mathbb{E}[q'(G)\Phi(\tau G)],$$

where we have used the evenness of $q$ to write $\mathbb{E}[q'(G)] = \mathbb{E}[Gq(G)] = 0$. We deduce that

$$a = 2\sigma^{-1}\mathbb{E}[q(G)\varphi(\tau G)], \quad b = 2\mathbb{E}[q'(G)\Phi(\tau G)]. \tag{74}$$

Lets write $C^{-1/2}c = a\bar{w}_g + b\bar{w}_o = \tilde{\beta}u + \beta v$, where $u = \bar{w}_o$ and $v$ is an unit-vector perpendicular to $u$ but in the plane spanned by $\bar{w}_o$ and $\bar{w}_g$. It is easy to see that

$$v = \frac{\bar{w}_g - \rho_g u}{\|\bar{w}_g - \rho_g u\|} = \frac{\bar{w}_g - \rho_g u}{\sqrt{1 - 2\rho_g^2 + \rho_g^2}} = \frac{\bar{w}_g - \rho_g u}{\sigma}.$$

We deduce that

$$\beta = c^\top v = (\bar{w}_g^\top v)a = \sigma a = 2\mathbb{E}[q(G)\varphi(\tau G)] =: \beta_2, \tag{75}$$

$$\tilde{\beta} = c^\top u = b + \rho_g a = 2\mathbb{E}[q'(G)\Phi(\tau G)] + 2\tau\mathbb{E}[q(G)\varphi(\tau G)]. \tag{76}$$

To match the formulae for $\beta_1$ and $\beta_2$ given in Table 3, we must now show that $\mathbb{E}[q'(G)\Phi(\tau G)] = \mathbb{E}[q(G)\Phi(\tau G)G] - \tau\mathbb{E}[q(G)\varphi(\tau G)]$ and conclude that $\tilde{\beta} = \beta_1$. To this end, write $\mathbb{E}[q'(G)\Phi(\tau G)] = \mathbb{E}[q'(G)f(G)]$, where $f(t) := \Phi(\tau G)$. By Stein's lemma (Gaussian integration by parts), we have

$$\mathbb{E}[q'(G)f(G)] = \mathbb{E}[q(G)(Gf(G) - f'(G))] = \mathbb{E}[q(G)(G\Phi(\tau G) - \tau\varphi(\tau G))]$$
$$= \mathbb{E}[q(G)\Phi(\tau G)G] - \tau\mathbb{E}[q(G)\varphi(\tau G)],$$

as claimed.

**Computing $p$ and $\gamma$.** We now compute the pruning ratio by definition as $p := \mathbb{E}[p_i] = \mathbb{E}[q(V)] = \mathbb{E}[q(G)]$ and $\gamma = \mathbb{E}[(x_i^\top w_o)^2 p_i] = \mathbb{E}[q(V)V^2] = \mathbb{E}[q(G)G^2]$ for $G \sim \mathcal{N}(0,1)$. This matches the formulae given in the first row of Table 3. $\qquad\square$

### L.2 LIMO Case

Let us now prove the formula for $\beta_1$ and $\beta_2$ given in the second row of Table 3. Here $F := \text{sign}(U)q(V)H(UV)$, where $H$ is the Heaviside step function with the convention $H(0) = 1/2$. Now, one computes

$$\frac{\partial F}{\partial U} = 2\delta(U)q(V)H(UV) + \text{sign}(U)q(V)V\delta(UV), \tag{77}$$

$$\frac{\partial F}{\partial V} = \text{sign}(U)q'(V)H(UV) + \text{sign}(U)q(V)U\delta(UV),$$
$$= \text{sign}(U)q'(V)H(UV) + |U|q(V)\delta(UV) \tag{78}$$

**Computing the $a$ coefficient.** One computes

$$\mathbb{E}[\delta(U)q(V)H(UV)] = \varphi(0)\mathbb{E}[\delta(U)q(V)H(0) \mid U = 0] = \frac{\varphi(0)}{2}\mathbb{E}[q(V) \mid U = 0]$$
$$= \ldots = \frac{1}{2\sigma}\mathbb{E}[q(G)\varphi(\tau G)].$$

On the other hand, using the well-known identity

$$\delta(xy) = \delta(y)/|x| + \delta(x)/|y|,$$

one computes

$$\mathbb{E}[\text{sign}(U)q(V)V\delta(UV)] = \mathbb{E}[\text{sign}(U)q(V)V\delta(V)/|U|] + \mathbb{E}[\text{sign}(U)q(V)V\delta(U)/|V|]$$
$$= \mathbb{E}[(1/U)q(V)\underbrace{V\delta(V)}_{=0}] + \mathbb{E}[\text{sign}(U)\delta(U)\,\text{sign}(V)q(V)]$$
$$= \varphi(0)\mathbb{E}[\text{sign}(V)q(V) \mid U = 0] = 0,$$

where the last step is because $t \mapsto \text{sign}(t)q(t)$ is an odd function, and the distribution of $V$ conditioned on $U = 0$ is $\mathcal{N}(0, \sigma^2)$ which is symmetric around the origin. We deduce that

$$a = \sigma^{-1}\mathbb{E}[q(G)\varphi(\tau G)]. \tag{79}$$

**Computing the $b$ coefficient.** For any $s \in \mathbb{R}$,

$$\mathbb{E}[\text{sign}(U)q'(V)H(UV) \mid V = s]$$
$$= q'(s)\varphi(s)\mathbb{E}[\text{sign}(U)1_{sU \geq 0} \mid V = s]$$
$$= q(s)\varphi(s)\left(\mathbb{P}(U \geq 0,\, sU \geq 0 \mid V = s) - \mathbb{P}(U < 0,\, sU \geq 0 \mid V = s)\right).$$

Now, since the distribution of $U$ conditioned on $V = s$ is $\mathcal{N}(\rho_g s, \sigma^2)$, we have

$$\mathbb{P}(U \geq 0,\, sU \geq 0 \mid V = s) = \begin{cases} \mathbb{P}(U \geq 0 \mid V = s) = \Phi(\tau s), & \text{if } s \geq 0, \\ \mathbb{P}(U = 0 \mid V = s) = 0, & \text{if } s < 0, \end{cases}$$

$$\mathbb{P}(U < 0,\, sU \geq 0 \mid V = s) = \begin{cases} \mathbb{P}(U < 0,\, U \geq 0 \mid V = s) = 0, & \text{if } s \geq 0, \\ \mathbb{P}(U < 0 \mid V = s) = \Phi(-\tau s), & \text{if } s < 0. \end{cases}$$

Therefore, $\mathbb{E}[\text{sign}(U)q'(V)H(UV) \mid V = s] = q'(s)\,\text{sign}(s)\varphi(s)\Phi(\tau|s|)$, and we conclude that
$$\mathbb{E}[\text{sign}(U)q'(V)H(UV)] = \mathbb{E}[q'(G)\Phi(\tau|G|)\,\text{sign}(G)],$$

with $G \sim \mathcal{N}(0, 1)$. Define $h(t) := \Phi(\tau|t|)\,\text{sign}(t)$. It is clear that
$$h'(t) = 2\delta(t)\Phi(\tau|t|) + \tau\varphi(\tau|t|) = 2\delta(v)\Phi(0) + \tau\varphi(\tau t) = \delta(v) + \tau\varphi(\tau t).$$

Gaussian integration by parts then gives
$$\mathbb{E}[q'(G)\Phi(\tau|G|)\,\text{sign}(G)] = \mathbb{E}[q'(G)h(G)] = \mathbb{E}[q(G)(Gh(G) - h'(G))]$$
$$= \mathbb{E}[q(G)\Phi(\tau|G|)|G|] - \tau\mathbb{E}[q(G)\varphi(\tau G)] - \varphi(0)q(0).$$

But $q'$ is odd (because $q$ is even), and also $t \mapsto \Phi(\tau|t|)$ is obviously even. We deduce that $\mathbb{E}[\text{sign}(U)q'(V)H(UV)] = 0$. Likewise, using the identity $\delta(UV) = \delta(V)/|U| + \delta(U)/|V|$, one computes

$$\mathbb{E}[|U|q(V)\delta(UV)] = \mathbb{E}[q(V)\delta(V)] + \mathbb{E}[|U|q(V)\delta(U)/|V|]$$
$$= \varphi(0)q(0) + \mathbb{E}[\underbrace{|U|\delta(U)}_{0}\, q(V)/|V|] = \varphi(0)q(0).$$

We deduce that $b = \mathbb{E}[q(G)\Phi(\tau|G|)|G|] - \tau\mathbb{E}[q(G)\varphi(\tau G)]$. Therefore, writing $C^{-1/2}c = \tilde{\beta}u + \beta v$ as before, we have

$$\beta = \sigma a = \mathbb{E}[q(G)\varphi(\tau G)] =: \beta_2,$$
$$\tilde{\beta} = b + \rho_g b = \mathbb{E}[q(G)\Phi(\tau|G|)|G|] =: \beta_1,$$

which are precisely the formulae given in Table 3.

**Computing $p$ and $\gamma$.** We now compute the pruning ratio $p := \mathbb{E}[p_i] = \mathbb{E}[q(V)H(UV)]$ and $\gamma := \mathbb{E}[(x_i^\top w_o)^2 p_i] = \mathbb{E}[V^2 q(V)H(UV)]$ by definition of $p_i$ in Eqn (6). Now, for any $s \in \mathbb{R}$, we have

$$\mathbb{E}[H(UV) \mid V = s] = \begin{cases} \mathbb{P}(U \leq 0 \mid V = s) = \Phi(-\tau s), & \text{if } s < 0, \\ 1/2, & \text{if } s = 0, \\ \mathbb{P}(U \geq 0 \mid V = s) = \Phi(\tau s), & \text{if } s > 0 \end{cases}$$
$$= \Phi(\tau|s|).$$

Integrating out $s$ with density $\varphi(s)$, we deduce that

$$p = \mathbb{E}[q(G)\Phi(\tau|G|)], \quad \gamma = \mathbb{E}[q(G)\Phi(\tau|G|)G^2],$$

as claimed. $\square$

# M    ANALYTIC FORMULAE FOR $p(q)$, $\gamma(q)$, $\beta(q)$, AND $\tilde{\beta}(q)$

Note that every symmetric pruning function $q \in \mathcal{Q}$ is the support function of sum $T := -S \cup S$, where $S$ is (up to a null set) a countable union of closed intervals. We consider a subclass of symmetric pruning functions corresponding to finite unions, i.e

$$q = 1_T, \text{ with } T = -S \cup S,\ S = \cup_{j=1}^{k}[a_j, b_j],\ 0 \leq a_1 < b_1 < a_2 < \ldots < a_k < b_k \leq \infty. \quad (80)$$

The "keep easy examples" (KE) and "keep hard examples" (KH) pruning functions used in (Sorscher et al., 2022) and defined defined below belong to this class $k = 1$ (for some $\alpha > 0$):

$$q_{\mathrm{KE}}(t) := 1[|t| \geq \alpha], \text{ i.e } q_{\mathrm{KE}}(t) = 1 \text{ if } |t| \geq \alpha \text{ and } q_{\mathrm{KE}}(t) = 0 \text{ otherwise,} \tag{81}$$

$$q_{\mathrm{KH}}(t) := 1[|t| \leq \alpha], \text{ i.e } q_{\mathrm{KH}}(t) = 1 \text{ if } |t| \leq \alpha \text{ and } q_{\mathrm{KH}}(t) = 0 \text{ otherwise,} \tag{82}$$

where $\alpha > 0$ which controls the proportion $p = \mathbb{E}[p_i]$ of training data which survives the curation.

Since they correspond to taking $S = [\alpha, \infty]$ and $S = [0, \alpha]$ respectively. The representation 80 also generalizes the setup of Feng et al. (2025) and Firdoussi et al. (2024) corresponds to $q \equiv 1$, i.e $S = [0, \infty]$.

For any $\alpha \in [0, \infty]$, define $I_k(\alpha) := \int_0^\alpha f_k(x)\varphi(x)\mathrm{d}x$, where the functions $f_k$ are defined by

$$f_1(x) := \Phi(\tau x), \quad f_2(x) := \varphi(\tau x), \quad f_3(x) := x\Phi(\tau x), \quad f_4(x) := x^2\Phi(\tau x).$$

As usual, $\varphi$ and $\Phi$ are the standard normal pdf and cdf respectively.

**Proposition 5.** *Consider a symmetric pruning function $q$ of the form Eqn (80).*

*(A) For label-agnostic curation Eqn (5), it holds that*

$$p(q) = \sum_{j=1}^{k} g(b_j) - g(a_j), \text{ with } g(z) := 2\Phi(z) - 1 \tag{83}$$

$$\gamma(q) = \sum_{j=1}^{k} g(b_j) - g(a_j), \text{ with } g(z) := 2(\Phi(z) - z\varphi(z)) - 1, \tag{84}$$

$$\beta_1(q) = ..., \tag{85}$$

$$\beta_2(q) = 2\varphi(0)\sigma \sum_{j=1}^{k} \Phi(b_j/\sigma) - \Phi(a_j/\sigma). \tag{86}$$

*(B) For Label-aware curation Eqn (6), it holds that*

$$p(q) = 2\sum_{j=1}^{k} I_1(b_j) - I_1(a_j), \tag{87}$$

$$\gamma(q) = 2\sum_{j=1}^{k} I_4(b_j) - I_4(a_j), \tag{88}$$

$$\beta_1(q) = 2\sum_{j=1}^{k} I_3(b_j) - I_3(a_j), \tag{89}$$

$$\beta_2(q) = 2\sum_{j=1}^{k} I_2(b_j) - I_2(a_j). \tag{90}$$

Part (A) of the proof follows directly from Eqn (8). Part (B) of the proof is a consequence of the identity $\int_a^b h(x)\mathrm{d}x \equiv I(b) - I(a)$, where $I(\alpha) := \int_0^\alpha h(x)\mathrm{d}x$, combined with the following lemma.

**Lemma 8.** *For any $\alpha \in [0, \infty)$, the following identities hold:*

$$I_1(\alpha) = \Phi(\alpha) - 1/2 - [\Phi_2(\alpha, 0; \rho) - \Phi_2(0, 0; \rho)], \tag{91}$$

$$I_2(\alpha) = \sigma\varphi(0)[\Phi(\alpha/\sigma) - 1/2], \tag{92}$$

$$I_3(\alpha) = \tau I_2(\alpha) - [\varphi(\alpha)\Phi(\tau\alpha) - \varphi(0)/2], \tag{93}$$

$$I_4(\alpha) = I_1 - \alpha\varphi(\alpha)\Phi(\tau\alpha) + \rho\sigma\left[\varphi(0)^2 - \varphi(\alpha)\varphi(\tau\alpha)\right]. \tag{94}$$

*The results are extended to $\alpha = \infty$ by noting that*

$$\lim_{\alpha\to\infty} \alpha\varphi(\alpha) = \lim_{\alpha\to\infty} \varphi(\alpha) = 0, \quad \lim_{\rho\to 1} \tau I_2(\alpha) = \frac{\varphi(0)}{2}.$$

