# OpenReview forum: "Why Less is More (Sometimes): A Theory of Data Curation"
_ICLR.cc/2026/Conference — ICLR 2026 Poster_

### Official Review · Reviewer_dewW · 2025-10-29

**Soundness:** 4
**Presentation:** 4
**Contribution:** 3
**Rating:** 8
**Confidence:** 2

**Summary:**

This paper introduces a principled theoretical framework for data curation, trying to figure out when less data actually works better than more, to challenge the classical "more is more" scaling laws with comprehensive experiments. It derives exact scaling laws for test error under label-agnostic and label-aware curation rules, showing that strategically pruned small datasets can outperform full datasets under conditions like strong generators and abundant data, while also mitigating model collapse.

**Strengths:**

1.  The first paper tries to theoretically explain when and why LIMO happens in machine learning by developing a principled theoretical framework for data curation, enabling practitioners to design data-centric pipelines instead of blindly scaling datasets.
2. Also provides a unifying explanation for contradictory curation strategies.
3. The writing and presentation look perfect to me.

**Weaknesses:**

1. While the paper links theory to LLM math reasoning (AIME benchmark in Table 1&2), it would be better to discuss other LLM-critical tasks (e.g., code generation, natural language understanding, or multi-step reasoning benchmarks) where curation is also widely used. Additionally, the authors do not investigate how the pruning ratio or oracle quality affects performance across different model sizes (e.g., 7B vs. 70B LLMs), leaving uncertainty about whether the LIMO principle holds consistently across LLM scales.

**Questions:**

1. The theory is framed around binary classification, but multi-class tasks are definitely more common in practice. Could you discuss whether the optimal pruning strategies could extend to multi-class settings?
2. In the ImageNet experiment, the generator and pruner are the same pre-trained ViT, which ensures alignment, but real-world CV pipelines may use separate generators (e.g., a synthetic data model) and pruners (e.g., a human-in-the-loop verifier). Could you discuss how misalignment between generator and pruner (e.g., low ρ₊ relative to ρ) would impact the optimal curation strategy, and whether this differs from the LLM case (where generator and pruner are often separate)?
3. I have another question for the authors, but not necessarily to be answered: LLM training encompasses pre-training, post-training, and stages such as SFT and RL within the fine-tuning process, with distinct training data used for each stage. For the LIMO method discussed in the paper, in which of these training stages is its principle more likely to be reflected?

---

> ### Author Response · Authors · 2025-11-21
> **Response to Reviewer dewW**
>
> **Extending to other LLM tasks and model scales.** We thank the reviewer for the comments regarding the scope of the tasks. Regarding the applicability to other tasks (coding, etc), we have a discussion in section 5 (related work) where we discuss data selection in domains beyond math, citing code generation with natural verifiers. We reference Haluptzok et al. (2022) regarding synthesized code and Ulmer et al. (2024) regarding dialogue agents. We believe extending our empirical benchmark across these additional tasks as a valuable avenue for future research to further expand on our theoretical findings, as noted in our "Future Directions" (Section 6).
>
> Regarding the model size, we would like to highlight that based on our theory, it is the strength of the generator (i.e how well its responses align with the ground truth) and not necessarily its raw size that allows us to do LIMO type data curation. Our theory predicts that for strong models (analogous to 70B), the keep hard strategy should be optimal. While for weaker models (analogous to 7B or difficult tasks), keep easy minimizes the error, supporting “more is more”.
>
> In summary, our framework suggests that the LIMO principle is not directly related to the model size, rather to its capability relative to the task difficulty.
>
> **Extending the theory to multiclass settings.** The reviewer raises a good point, which is already hinted to, in the conclusion of our manuscript. We believe the optimal strategies continue to carry over to multi-class settings, as shown by our experiments on multi-class problems (classification, math reasoning, etc.). The theoretical analysis will be slightly more difficult: one would need to analyse the cross-entropy loss instead of logistic or squared loss (as done in our current analysis). The tools of random matrix theory would still be applicable here, the theoretical results will be considerably more complicated, as they would be given by fixed-point equations involving proximal operators, etc. This will be done in future work.
>
> **Alignment versus misalignment of generator and pruning oracle.** In our ImageNet experiments, the generator and pruner are the same pre-trained ViT model, which means the pruning direction is aligned with the generator’s labeling function. This setup simplifies the analysis and ensures that the pruner is highly effective at selecting informative and correctly labeled examples. However, in real-world scenarios, the generator (e.g., a synthetic data model) and the pruner (e.g., a human verifier or a separate model) are often distinct. This introduces the possibility of misalignment between the generator’s labeling function and the pruner’s selection criteria, which is captured by a lower value of $\rho$ (the cosine similarity between the pruner and the generator, defined in Eqn xyz).
>
> **Is LIMO pre-training or or post-training?** Pre-training typically uses vast, noisy, and diverse data, with little or no label information. LIMO-style pruning is difficult to apply at scale due to lack of ground-truth or reward signals. SFT uses labeled data, often curated for quality and diversity. LIMO’s principle aligns well here: one can prune the fine-tuning set to keep only the hardest, most informative, or most reliable examples (e.g., high-entropy, challenging instructions, or verified answers). Finally, RLHF involves iterative feedback, reward models, and adaptive data selection. LIMO’s principle could guide which examples to present for human feedback or which trajectories to reinforce, especially by focusing on challenging or high-reward cases. The theory may need to be extended to account for adaptive or evolving oracles.

---

### Official Review · Reviewer_tAw9 · 2025-10-31

**Soundness:** 4
**Presentation:** 3
**Contribution:** 3
**Rating:** 8
**Confidence:** 4

**Summary:**

The paper develops a teacher–student, high-dimensional model of learning from curated data in order to reconcile two empirically observed regimes in modern LLM/vision training: "More is more" regime (classical scaling laws) in Kaplan et al. (2020) and Hoffmann et al. (2022), where performance improves monotonically with dataset size vs. "Less is more" regime (LIMO/s1–style curation), when the generator is strongly aligned with the target, then selectively keeping only the most informative/hard examples yields strictly lower test error than training on the full set. Formally, the authors setup a theoretical model with Gaussian inputs and labels generated by a linear teacher with known alignment, and the learner as a ridge (square-loss) estimator trained on a pruned sample whose selection is governed by an oracle with its own alignment parameter. Using random-matrix/deterministic-equivalent techniques, the authors derive closed-form high-dimensional limits for the test error after curation. The authors show numerical simulations of their theory and corroborated with empirical simulations.

**Strengths:**

* The paper is clearly written and well-organized. The paper has both solid theoretical side and empirical evidence.
* The paper has deep theoretical building blocks with high dimensional asymptotics of linear models and classification problems. Using both RMT techniques and tools from Feng et al. for classification problems to study the effects of pruning by the oracle. The conclusions are sound as far as I have checked, and the results appear novel to me.
* The paper's conclusion provides an alternative to neural scaling laws: it is possible to perform with higher accuracy with fewer data with an expert pruner. This also gives a potential remedy to the neural collapse phenomenon where the training procedure seeks to scale with self-generated data---perhaps by scrutiny using the synthetic data (instead of training on them) and picking only harder/easier problems we can avoid model collapse from synthetic data.

**Weaknesses:**

Major comments:
* I think the main inconsistency between the main message and theory is that: both "KE" and "KH" strategies are effectively "less is more". It is just what data should be picked. I think it is a bit of stretch to say "KE" is equivalent to "more is more", while the neural scaling laws scenario should correspond to a very small $\phi$ instead of the "KE" strategy.
* The theory relies on linear model with Gaussian covariates, and is limited to the binary classification setup with proportional asymptotics. This is, definitely limited, as is acknowledged also by the authors in the limitation discussions, a good starting point for study the pruning and data curation in large datasets. I think this is only a minor weakness.

Minor comments:
* A general style suggestion: use \eqref instead of \ref for equations.
* Page 19, "Corollary xyz". I think this is a placeholder or typo.

**Questions:**

* My main questions are in the weaknesses section. I would especially like to see more discussions between "KE" and "more is more".
* The paper uses deterministic equivalence to obtain risk asymptotics for the problem (2). While in general, when l is a convex loss function, using the CGMT techniques the similar asymptotics can also be obtained. See Karoui (2013). What do authors think the behavior in this setup? Would the same behavior also be expected?

Karoui, Noureddine El. "Asymptotic behavior of unregularized and ridge-regularized high-dimensional robust regression estimators: rigorous results." arXiv preprint arXiv:1311.2445 (2013).

---

> ### Author Response · Authors · 2025-11-21
> **Response to Reviewer tAw9**
>
> **Clarifying LIMO vs MIMO, in relation to KE and KH.** The reviewer is probably referring to the paragraph just after Theorem 2 in the manuscript. The phrasing of that paragraph is indeed slightly confusing, and has been fixed in the update manuscript (changes are depicted in magenta color). The paragraph now reads as follows: *Part (A) shows that for a strong model/generator that has already mastered the task, performance is refined by focusing on difficult examples—a ”less is more” (Ye et al., 2025) approach. Part (B) captures the opposite scenario: for a weak model/generator, the best strategy is to keep easy examples. This latter case is particularly relevant for mitigating model collapse, where a model trained on its own imperfect outputs acts as a poor generator (Shumailov et al., 2024; Dohmatob et al., 2025). Also see Appendix C.*
>
>
> Finally, taking our theoretical results and empirical evidence together, we can boldly put forth the following take-home message: "Less is More" corresponds to KH + small $p$ (i.e. aggressive pruning and keep only hard examples). This is optimal when the alignment $\rho$ of the generator relative to the ground truth is high (i.e., strong generator). "More is More" corresponds to large $p$ (i.e., $p$ tending to 1). This setting is relevant when $\rho < 1$ (weak generator); one needs to use all the data we can get, so as to build a foundational understanding of the underlying task.
>
>
> **Extension to general convex loss functions.** The reviewer makes a great point. Indeed, the Convex Gaussian Min-Max Theorem (CGMT) is a natural idea for extending our results to the case of general convex loss functions. This has been done in Karoui (2013) (Albeit, from first principles) and even more recently. Yet another promising RMT-based route for analysing high-dimensional estimators has been developed in Xiaoyi Mai et al. (2019) "A Large Scale Analysis of Logistic Regression: Asymptotic Performance and New Insights". Applied to our setup, there is a significant technical hurdle, handling the effect of data selection, i.e., the weights $q_i$. The resulting analysis would likely lead to complicated fixed-point equations involving the proximal operator of the loss function, and the limiting spectral density of the covariance matrices $C$ and $\Sigma$. At any rate, such an analysis is completely feasible with additional effort, and will be considered in future work.
>
>
> **Typos and formatting.** We apologize for typos which have been corrected in the updated version of the manuscript (refer to text pieces colored magenta).
> - Typo on page 19: By “... Corollary xyz…”, we meant “Note that when $p=1$ (corresponding to no pruning), the above result recovers one of the main results of Dohmatob et al. (2025), namely, their Corollary 1.”
> - Referencing equations: We have also switched from \ref to \eqref all through the manuscript, as suggested by the reviewer.

---

> > ### Comment · Reviewer_tAw9 · 2025-11-25
> > **Response to the revision**
> >
> > Thank you for the thoughtful responses and revision! I remain my positive evaluation of the paper, and good luck!

---

### Official Review · Reviewer_CgiU · 2025-11-01

**Soundness:** 3
**Presentation:** 2
**Contribution:** 3
**Rating:** 6
**Confidence:** 2

**Summary:**

This work seeks to develop a theoretical framework for understanding data curation. In other words, they seek a theory that can predict when it is best to use less data (filtered/curated) versus all data (no filtering/curation). The authors are motivated by the fact that there are cases where small datasets have been shown to be beneficial, like LIMO and s1. Their theoretical results assume a Gaussian feature model under a binary classification setting.

**Strengths:**

1. The authors use their theory to show that it can predict empirical results to a surprising degree (mean relative error of 1.8% according to App. B), and within these empirical results live the settings they sought to understand: when all data is needed ("more is more") and when curation is needed ("less is more")
2. Authors consider both label-agnostic and label-aware data curation in their theory
3. Significant extra detail for figures and proofs are provided in appendix, answering a few of my early questions

**Weaknesses:**

1. Arguably the most important feature of the paper is that their theory is predictive of empirical results in a very realistic dataset (ImageNet training of a ViT), but it is not clear from the writing how to use this in practice. In particular: how does someone measure the quality of the generator $\rho$ for a dataset you are given? No commentary on this is given. A bonus would be if the authors gave a step-by-step appendix section for a practitioner on how to use their theory (willing to raise score for clarity on this)
2. "the base LLM is a strong generator" based on what? how is this claim measured?
3.  Why is $n=5000$ considered the "large data" setting in Figure 1? How was this number chosen? That $n$ is smaller than the CIFAR-10 training set...

Minor:
- capitalize PDF and CDF on L184

**Questions:**

1. Why is the Gaussian feature model noted as a limitation? Wouldn't we expect gaussian distributions of features according to central limit theorem? The authors mention real-world data is "structured" but I find that vague.
2. How does someone measure the quality of the generator $\rho$ for a dataset you are given?
3. Is it possible to know the quality of pruning? For example, suppose I curate a dataset of text by removing duplicates, how would a "quality" be assigned to that?

---

> ### Author Response · Authors · 2025-11-21
> **Response to Reviewer CgiU**
>
> **Why is the Gaussian feature model noted as a limitation?** In real life, the distribution of features is very far from being Gaussian. Fortunately, the Gaussianity assumptions can be weakened to: finite 2nd order moments, and sub-Gaussian concentration. Assuming Gaussianity just greatly simplifies the analysis.
>
> Assuming Gaussianity greatly simplifies the analysis. The random matrix theory (RMT) part of our analysis is robust to the Gaussianity assumption. It is enough that the distribution of the features have finite second order moments and have a sub-Gaussian concentration profile.
>
> This Gaussianization carries over to the case of non-linear models provided there are in the so-called NTK regimes (e.g., if they are very wide). The second order statistics of the induced random features can be captured via Gaussian random matrices. Such a connection would allow our analysis to extend from linear to non-linear regimes (as we stay in the said NTK regime).
> Finally, note that the statistical quantities whose analysis must be analyzed in our theory necessitate more than just central limit theorems (CLTs). It’s possible we have misunderstood what the reviewer intended to say here. We are happy to cycle back on this point.
>
> **Estimating oracle quality.** In our framework, the “quality” of a pruning/curation rule is not a subjective score but the (implicit) agreement of that rule with an oracle that knows which examples are correct and informative. Formally, this is encoded by the confusion matrix of the pruner w.r.t. the oracle (true-positive / false-positive rates), which is exactly what our quality parameters summarize in the scaling laws. In practice, for a heuristic such as de-duplication of text, we do not know these quantities analytically, but they can be **estimated** or **proxied**:
> - on a small labeled subset, by measuring how often the pruner keeps “good” examples and removes “bad” ones (off-topic, toxic, low-information, etc.);
> - by fitting our scaling-law curves to a family of datasets obtained by varying the intensity of the same curation rule (e.g., different de-dup thresholds) and reading off the corresponding effective quality parameter;
> - or, in the spirit of (Feng et al., 2024), via a task-specific similarity/verification score (e.g., ROUGE, math-verifier accuracy) whose derived proxy is empirically strongly correlated with downstream performance.
>
> So, in summary: yes, even for simple rules like duplicate removal, our theory says that “quality” is whatever scalar (or small set of parameters) best summarizes how the pruner’s decisions align with an oracle, and this can be **estimated from data** rather than assumed. We will clarify this in the text with an explicit example (e.g., de-duplication) and a short discussion of such proxy-based estimation.
>
> **Estimating generator quality.** Aggressive pruning is only effective when the model effectively “knows” the easy data. A strong generator is one that has mastered the easy data and therefore benefits from high-quality, informative (hard) samples to refine subtle deviations. For example, in Section 4.2, we report results on “average” and “hard” slices of the AIME benchmark. For the majority of AIME problems, modern LLMs (like Qwen 2.5) act as strong generators, so the “keep hard” (LIMO) strategy works best. However, on the 'hard' slice of the data, these same LLMs act as “weak generators”. Consequently, aggressive pruning is not beneficial; instead, a 'more is more' approach is required to build the necessary foundational skills for these difficult problems.
>
> **Wording.** Indeed our use of the term “structured” is unclear, and will be replaced by “generally non-Gaussian”.

---

### Official Review · Reviewer_UCRG · 2025-11-01

**Soundness:** 3
**Presentation:** 4
**Contribution:** 3
**Rating:** 8
**Confidence:** 3

**Summary:**

This paper introduces a rigorous theoretical framework to analyze the efficacy of data curation strategies in high-dimensional learning. It addresses a central paradox in modern ML: When does aggressive data pruning ("Less is More," exemplified by methods like LIMO and s1) outperform training on the full dataset ("More is More")?

The paper model this scenario using high-dimensional binary classification and regression with Gaussian covariates. The setup captures the interactions between a data generator ($w_g$), the ground truth ($w_*$), and a pruning oracle ($w_o$), allowing for label shift. They analyze both label-agnostic (based on difficulty/margin) and label-aware (based on difficulty and correctness) curation.

Using RMT, the paper derive exact asymptotic expressions for the test error in the proportionate scaling limit ($d/n \to \phi$). The central theoretical finding (Theorem 2) identifies a precise phase transition: "Less is More" (specifically, keeping hard examples) is optimal iff data is abundant AND the generator is strong. In all other regimes (scarce data or weak generator), using more data or keeping easy examples is superior.

While the paper is primarily theoretical, it tries rounding itself off with some simulations and Imagenet experiments.

**Strengths:**

- **Theoretical Rigor and Exactness:** The primary strength is the sophisticated mathematical analysis. By leveraging RMT, the paper derive *exact* asymptotic formulas for the generalization error (Theorems 1, 3). This allows for a precise characterization of the interplay between data quality ($\rho$), oracle quality ($\rho_*$), and data scale ($\phi$), moving far beyond bounds or heuristics. The derivation of how pruning "deforms" the Marchenko-Pastur law governing the data spectrum is technically impressive and commendable.

- **Good Resolution of the "Less vs. More" question:** The paper provides a clear and elegant resolution to conflicting empirical observations regarding data scaling. Fig. 1 beautifully illustrates the four key regimes of learning. The insight that the optimal strategy is fundamentally tied to the generator's strength relative to the task difficulty is a major conceptual contribution.

- The framework successfully unifies several recent lines of work, including data pruning strategies and model collapse mitigation. The analysis of label-aware curation (Section 3.2) is highly relevant to modern SFT/RL*F pipelines where reward models filter policy outputs. The demonstration that strategic pruning prevents model collapse (Fig. 3) is practically significant for the stability of iterative training loops.

- The authors provide some practical validation. Synthetic experiments show strong agreement with the theory (App. B, Fig. 4). Crucially, the ImageNet experiments confirm that the qualitative insights transfer to deep learning settings (ViT), and the analysis in Sec. 4.2 provides a compelling explanation for recent contradictory LLM results.

**Weaknesses:**

**Linear Models:** The theoretical results are derived under the assumption of linear models and Gaussian covariates. This is a known standard simplification for RMT analysis but limits the direct quantitative applicability to deep neural networks on structured data. However, the empirical results do point towards potential qualitative transfer, but would warrant a more complete exploration.

**Fixed Oracle:** The framework assumes a fixed pruning oracle ($w_o$). In practice, oracles (e.g., reward models) are often learned and may evolve adaptively during training (e.g., iterative RLHF), which is not captured in the current analysis. Extending the theory to adaptive oracles would be an important future direction.


I am **highly skeptical** that this constitutes the explanation for "Less is More" in modern deep learning. The assumptions are too strong, the dynamics of optimization (like SGD) and feature learning are ignored, and the empirical validation is not sufficiently robust to confirm that the mechanisms identified by the theory are the ones driving results in practice.

**Questions:**

- There is an inherent balance/tradeoff between optimal strategy vs model collapse mitigation. Could the authors further clarify the contradictory observation between Theorem 2(b) and Fig.3 ? To be more specific, does the optimal strategy shift from Keeping easy back to Keeping Hard when moving from label-agnostic curation to label-aware curation? Maybe an equivalent Theorem 2 for label-aware setting could help.

- An important practical consideration is to identify which of the four regimes in Fig. 1 is one operating in. Practically where ground truth is scarse and strength of model is difficult to estimate, how would the paper suggest reliably diagnose this? I guess there needs to be a detector for this crossover point when KH > KE?

- The RMT analysis relies on linear models, while experiments elude to ImageNet teasing the qualitative transfer, but the theoretical boundaries remain tied. Can the paper describe the mechanisms they believe enable this transfer?

---

> ### Author Response · Authors · 2025-11-21
> **Response to Reviewer UCRG**
>
> **Extending the analysis beyond linear models.** Thank you for this comment. Our analysis can be easily extended to kernel methods with dot-product kernels, and thus to infinite-width neural networks. Furthermore, under the NTK hypothesis (whereby network parameters stay close to the initialization during training), our analysis can be extended to wide finite-width neural networks. Such an analysis can be carried out using the tools of operator-valued free probability theory. This direction was hinted at in the conclusion section of our manuscript. In such setups, the effect of activation function can be taken into account via its Gaussian norm and first 2 Hermite coefficients, thanks to the so-called Gaussian Equivalence Theorem (Goldt et al. 2022, *“The Gaussian equivalence of generative models for learning with shallow neural networks”*). Such an analysis will be carried in future work.
>
>
> Less obvious is extending the analysis to feature-learning regimes (e.g., SGD on moderately parametrized networks). Here, the problem becomes significantly more difficult and we can no longer rely on classical RMT. This is an interesting future direction.
>
>
> **Adaptive oracles.** Indeed, extending our theoretical analysis to account for adaptive oracles would indeed be a natural extension and a promising avenue for future research (as mentioned in the conclusion section of our manuscript). It should be possible to carry such an analysis with the same RMT ideas we have employed (appendix). One only needs to carefully account for the induced dynamics on the weights $p_i$ that appear in the objective $p_i$. Amongst other changes, this would induce dynamical versions of the constants in equations (7), (8), and (12).
>
>
> **Extending the analysis to more realistic setups.** The reviewer is correct to point out that our analysis doesn’t capture optimization dynamics which induce a rich feature learning regime (SGD, etc.). While we do not claim that our theory provides a complete or exclusive explanation for "Less is More" in all modern deep learning contexts, we believe it offers a valuable perspective and a concrete starting point for further investigation. Our empirical results, though not exhaustive, do suggest that the mechanisms highlighted by our analysis are relevant to the modern deep learning setup.
>
>
> **Reconciling Theorem 2(b) and Fig. 3.** Indeed, we don’t yet have a version of **Theorem 2(b)** for the label-aware data curation setup. Empirically, we observe that for label-aware data citation, the optimal pruning strategy depends on proportion $p$ retained data points: it can alternate between KE (“keep easy”) and KH (“keep hard”). A theoretical analysis of this scenario deserves further study. Such an analysis is considerably more difficult than the label-agnostic case and is left for future work.
>
>
> Not withstanding, we empirically observe in Figure 3 that the KH label-aware pruning strategy successfully mitigates model collapse. In view of the preceding paragraph above, it is at the moment not clear whether KH is optimal mitigation strategy

---

> > ### Comment · Reviewer_UCRG · 2025-11-26
> >
> > While acknowledging the shortcomings is okay, I would like the authors to explicitly acknowledge these in the paper to motivate further research. Besides this, I keep my score as the paper is still overall well done. Looking forward to its follow-ups.

---

### Author Response · Authors · 2025-11-21
**General Response to All Reviewers.**

We thank the reviewers for their useful insights. Below, we address the questions of each reviewer. We have also made minor changes to the manuscript (highlighted in magenta color).

---

### Meta-Review · Area_Chair_xPPB · 2026-01-05

**Summary:**

- The theoretical results are only derived under the assumption of linear models and Gaussian covariates, which limits the direct quantitative applicability to deep neural networks on structured data.

- Uses a fixed pruning oracle.

- The assumptions are too strong, the dynamics of optimization (like SGD) and feature learning are ignored, and the empirical validation is not sufficiently robust to confirm that the mechanisms identified by the theory are the ones driving results in practice.

- How to use the work in practice?

- It would be better to discuss other LLM-critical tasks.

**Reviewer Concerns:**

I think the concerns about the linear models, pruning oracle, and discussion are addressed. The concern about the application still holds. However, I believe the authors have made a persuasive rebuttal.

**Reviewer Scores:**

The four positive reviewers hold their scores.

---

### Decision · Program_Chairs · 2026-01-26

Accept (Poster)